# Combinations of *Spok* genes create multiple meiotic drivers in *Podospora*

**Aaron A Vogan[1†], S Lorena Ament-Velásquez[1†], Alexandra Granger-Farbos[2], Jesper Svedberg[1], Eric Bastiaans[3], Alfons JM Debets[3], Virginie Coustou[2], Hélène Yvanne[2], Corinne Clavé[2], Sven J Saupe[2], Hanna Johannesson[1]\***

[1]Organismal biology, Uppsala University, Uppsala, Sweden; [2]University of Bordeaux, Bordeaux, France; [3]Wageningen University, Wageningen, Netherlands

**Abstract** Meiotic drive is the preferential transmission of a particular allele during sexual reproduction. The phenomenon is observed as spore killing in multiple fungi. In natural populations of *Podospora anserina*, seven spore killer types (*Psk*s) have been identified through classical genetic analyses. Here we show that the *Spok* gene family underlies the *Psk*s. The combination of *Spok* genes at different chromosomal locations defines the spore killer types and creates a killing hierarchy within a population. We identify two novel *Spok* homologs located within a large (74–167 kbp) region (the *Spok* block) that resides in different chromosomal locations in different strains. We confirm that the SPOK protein performs both killing and resistance functions and show that these activities are dependent on distinct domains, a predicted nuclease and kinase domain. Genomic and phylogenetic analyses across ascomycetes suggest that the *Spok* genes disperse through cross-species transfer, and evolve by duplication and diversification within lineages.

DOI: https://doi.org/10.7554/eLife.46454.001

**\*For correspondence:**
hanna.johannesson@ebc.uu.se

[†]These authors contributed equally to this work

**Competing interests:** The authors declare that no competing interests exist.

## Introduction

The genomes of all Eukaryotes harbor selfish genetic elements that employ a variety of mechanisms to undermine the canonical modes of DNA replication and meiosis in order to bias their own transmission (*Werren et al., 1988*; *Burt and Trivers, 2009*). As the proliferation of these elements is independent of the regulated reproduction of the host organism, they can create conflict within the genome (*Rice and Holland, 1997*). Such intragenomic conflict is predicted by theory to spur an arms race between the genome and the elements and, consequently, to act as a major driver of evolutionary change (*Werren, 2011*). To understand the extent to which intragenomic conflict has shaped the evolution of genomes and populations, it is crucial to identify the selfish genetic elements that are able to impact the dynamics of natural populations.

One important class of selfish genetic elements are known as meiotic drivers. These elements use a variety of mechanisms to hijack meiosis in order to bias their transmission to the gametes in proportions greater than 50% (*Sandler and Novitski, 1957*). This segregation distortion of alleles can be difficult to observe unless it is linked to an obvious phenotype such as sex (*Sandler and Novitski, 1957*; *Helleu et al., 2014*), thus the prevalence of meiotic drive in nature is probably underestimated. Nevertheless, meiotic drive has been observed in many model systems, including *Drosophila*, *Mus*, *Neurospora*, and *Zea mays*, suggesting that it is widespread across all major Eukaryotic groups (*Lindholm et al., 2016*; *Bravo Núñez et al., 2018b*). In ascomycete fungi, meiotic drive occurs in the form of spore killing, which represents the most direct way to observe the presence of drive (*Turner and Perkins, 1991*). When a strain possessing a driving allele mates with a compatible strain that does not carry the allele (i.e., a sensitive strain), the meiotic products (ascospores) that carry the driving allele will induce the abortion of their sibling spores that do not have the allele. Spore killing is apparent in the sexual structures (asci) of the fungi because it results in half of the normal number

**eLife digest** In many organisms, most cells carry two versions of a given gene, one coming from the mother and the other from the father. An exception is sexual cells such as eggs, sperm, pollen or spores, which should only contain one variant of a gene. During their formation, these cells usually have an equal chance of inheriting one of the two gene versions.

However, a certain class of gene variants called meiotic drivers can cheat this process and end up in more than half of the sexual cells; often, the cells that contain the drivers can kill sibling cells that do not carry these variants. This results in the selfish genetic elements spreading through populations at a higher rate, sometimes with severe consequences such as shifting the ratio of males to females.

Meiotic drivers have been discovered in a wide range of organisms, from corn to mice to fruit flies and bread mold. They also exist in the fungus *Podospora anserina,* where they are called 'spore killers'. Fungi are often used to study complex genetic processes, yet the identity and mode of action of spore killers in *P. anserina* were still unknown.

Vogan, Ament-Velásquez et al. used a combination of genetic methods to identify three genes from the *Spok* family which are responsible for certain spores being able to kill their siblings. Two of these were previously unknown, and they could be found in different locations throughout the genome as part of a larger genetic region. Depending on the combination of *Spok* genes it carries, a spore can kill or be protected against other spores that contain different permutations of the genes. Copies of these genes were also shown to be present in other fungi, including species that are a threat to crops.

Scientists have already started to create synthetic meiotic drivers to manipulate how certain traits are inherited within a population. This could be useful to control or eradicate pests and insects that transmit dangerous diseases. The results by Vogan, Ament-Velásquez et al. shine a light on the complex ways that natural meiotic drivers work, including how they can be shared between species; this knowledge could inform how to safely deploy synthetic drivers in the wild.

DOI: https://doi.org/10.7554/eLife.46454.002

of viable spores. Owing to the haplontic life cycle of most fungi, spore killing is unusual among meiotic drivers as it is the only system in which the offspring of an organism are killed by the drive (*Lyttle, 1991*). In addition, with few exceptions (*Hammond et al., 2012*; *Svedberg et al., 2018*), spore killer elements appear to be governed by single loci that confer both killing and resistance (*Grognet et al., 2014*; *Nuckolls et al., 2017*; *Hu et al., 2017*), which contrasts with the other well-studied drive systems that comprise genomic regions as large as entire chromosomes (*Larracuente and Presgraves, 2012*; *Hammer et al., 1989*).

Meiotic drivers are often expected to reach fixation or extinction in populations relatively rapidly (*Crow, 1991*), at which point the effects of the drivers will no longer be observable. In agreement with this expectation, most drivers that have been described exhibit large shifts in frequencies in both time and space (*Lindholm et al., 2016*; *Carvalho and Vaz, 1999*). In the case of spore killers, multiple drivers have been found to coexist within a given species. The evolutionary dynamics of multiple drivers within species has not been thoroughly explored, but two contrasting examples are known. In the genomes of *Schizosaccharomyces pombe*, numerous copies of both functional and pseudogenized versions of the *wtf* driver genes are found (*Nuckolls et al., 2017*; *Hu et al., 2017*; *Eickbush et al., 2019*). By contrast, the two spore killers *Sk-2* and *Sk-3* of *Neurospora intermedia* have only been described in wild strains four times and once, respectively, whereas resistance to spore killing is widespread (*Turner, 2001*). The impact of multiple drivers coexisting in a single population was not characterized in either of these cases.

Natural populations of the filamentous fungus *Podospora anserina* are known to host multiple spore killers (*Grognet et al., 2014*; *van der Gaag et al., 2000*; *Hamann and Osiewacz, 2004*), and hence provide an ideal system for the investigation of interactions among drivers at the population level. The first spore killer gene to be described in *P. anserina* was *het-s*, a gene that is also involved in allorecognition (*Dalstra et al., 2003*). Another class of spore killer genes in *Podospora* are known as *Spok* genes. *Spok1* is only known from a single representative of *P. comata*, a species that is

closely related to *P. anserina*, whereas *Spok2* has been shown to exist in high frequency among strains of a French population of *P. anserina* (*Grognet et al., 2014*). *Spok1* is capable of killing in the presence of *Spok2*, but not vice versa, indicating a dominant epistatic relationship between the two genes. In addition, seven spore killer genotypes have been identified through classical genetic analysis (*van der Gaag et al., 2000*). These are referred to as *Psk-1* through *Psk-7* and were defined by observing the presence, absence and frequency of killed spores in defined crosses among French and Dutch *P. anserina* strains (*Box 1—figure 1*). At the onset of this study, it was not known whether the *Psk* elements represent independent meiotic drive genes, or whether they may be related to the *Spoks* and/or allorecognition loci. The *het-s* gene itself is not associated with the *Psks*, but allorecognition is correlated with *Psk* spore killing (*van der Gaag et al., 2003*). On the other hand, the relationship between the *Spoks* is reminiscent of the hierarchy of killing among the *Psks*, suggesting a possible connection between the activity of *Spok* genes and *Psks*.

The primary goal of this study was to determine the identity of the genes that are responsible for the *Psk* spore-killer types found in *P. anserina*, and whether they relate to known meiotic drive genes. We identified two novel *Spok* homologs (*Spok3* and *Spok4*) and showed that these two, together with the previously described *Spok2*, represent the genetic basis of the *Psk* spore killers. The new *Spoks* occur in large novel regions that can be found in different genomic locations in different strains. Our results illuminate the underlying genetics of a polymorphic meiotic drive system and expand our knowledge regarding their mechanism of action.

## Results

### Genome assemblies

To investigate the genetic basis of spore killing in *P. anserina*, we generated high-quality whole-genome assemblies using a combination of long-read (PacBio and MinION Oxford Nanopore) and short-read (Illumina HiSeq) technologies. *Table 1* lists the strains used for investigation. In all cases, we sequenced single haploid monokaryons (marked with + or - following the strain name, to designate their mating type; see 'Materials and methods' and Appendix 1). We selected strains from a natural population in Wageningen (Wa), the Netherlands, and a few strains from France, representing six of the seven previously described *Psk* spore killer types (*Psk-1*, *Psk-2*, *Psk-3*, *Psk-4*, *Psk-5* and *Psk-7*; *van der Gaag et al., 2000*) along with a strain of a novel killing type (Wa100), to which we assign the type *Psk-8*, and strain Wa63. The reference strain of *P. anserina*, S, was not given a *Psk* designation previously, as it was not known to induce spore-killing. However, *Grognet et al. (2014)* demonstrated that it can indeed induce spore-killing, so here we assign it to *Psk-S* along with Wa63. In addition, we acquired and sequenced strains from the closely related *Podospora* species *P. pauciseta* (CBS237.71) and *P. comata* (strain T). A strain annotated as T was acquired from two different laboratories, one from the Wageningen collection (referred hereafter as $T_D$) and one from Goethe University Frankfurt (here as $T_G$). Our results revealed that these strains do not represent the same isolate, as previously thought (*Hamann and Osiewacz, 2004*), but are distinct. $T_G$ is a *Psk-5* strain of *P. anserina* and was sequenced with Nanopore and Illumina. $T_D$ matches the *P. comata* epitype reported by *Silar et al. (2019)* and was sequenced with Illumina alone.

The final assemblies (long-read technologies polished with Illumina HiSeq data) recovered the expected seven chromosomes in their entirety for five strains, and in up to 13 scaffolds for the rest (*Supplementary file 1*). BUSCO analyses of these assemblies reported 97–98% of 3725 Sordariomyceta-conserved genes (*Supplementary file 1*), which is concordant with the same analysis done in the reference assemblies of the *P. anserina* strain S+ (hereafter referred to as Podan2; *Espagne et al., 2008*) and of *P. comata* (PODCO; *Silar et al., 2019*). Notice that as the assemblies of each strain were produced from one haploid (monokaryotic) isolate, we will refer to specific genome assemblies with their strain name followed by their corresponding mating type; for example, the assembled genome of monokaryon Wa63+ (derived from the strain Wa63) is called PaWa63p (*Supplementary file 1*).

In addition, all genomes sequenced with Illumina were assembled de novo using SPAdes. The resulting assemblies consisted of between 222 and 418 scaffolds that were larger than 500 bp, with a mean N50 of 227 kbp (*Supplementary file 2*). The alignment coverage of Podan2 (*Espagne et al., 2008*) was above 98% for all of the SPAdes assemblies of *P. anserina*. When the filtered Illumina

## Box 1. The seven separate *Psk*s are defined by their spore-killing percentage and mutual interactions.

To understand how the spore-killing percentages relate to the *Psk* types of the strains, it is necessary to first appreciate some of the fundamental aspects of *Podospora* biology. Within the fruiting body (perithecium), dikaryotic cells undergo karyogamy to produce a diploid nucleus that immediately enters meiosis. After meiosis, one round of post-meiotic mitosis occurs, resulting in eight daughter nuclei. The nuclei are packaged together with their non-sister nuclei from mitosis (dashed line) to generate dikaryotic, self-compatible spores. In a cross in which the parental strains harbor two alternative alleles for a given gene of interest (one of which is indicated by the red mark on the chromosome), spores that are either homoallelic or heteroallelic for the gene can be produced, depending on the type of segregation. Specifically, if there is no recombination event between the gene and the centromere, the gene undergoes first division segregation (FDS) and the parental alleles co-segregate during meiosis I, generating homoallelic spores (i). FDS of a spore-killing gene will thus result in a two-spored ascus (ii). If there is a recombination event between the gene of interest and the centromere, second division segregation (SDS) occurs. In this case, heteroallelic spores will be formed (i). For spore-killing, a four-spored ascus will still be produced because only one copy of the spore killer is required to provide resistance (ii). As SDS is reliant on recombination, the frequency of SDS relates to the relative distance from the centromere and can be used for linkage mapping. When there is spore-killing, the percentage of two-spored asci is the frequency of FDS, and is referred to as the 'spore-killing percentage'. The *Psk*s were described by crossing different strains and evaluating their spore-killing percentage in each cross. The seven unique *Psk*s were shown to interact in a complex hierarchy, showing either a dominance interaction, mutual resistance, or mutual killing. (See Appendix 1 for more details and *Figure 4—figure supplement 1* for a reproduction of the hierarchy presented in **van der Gaag et al., 2000**).

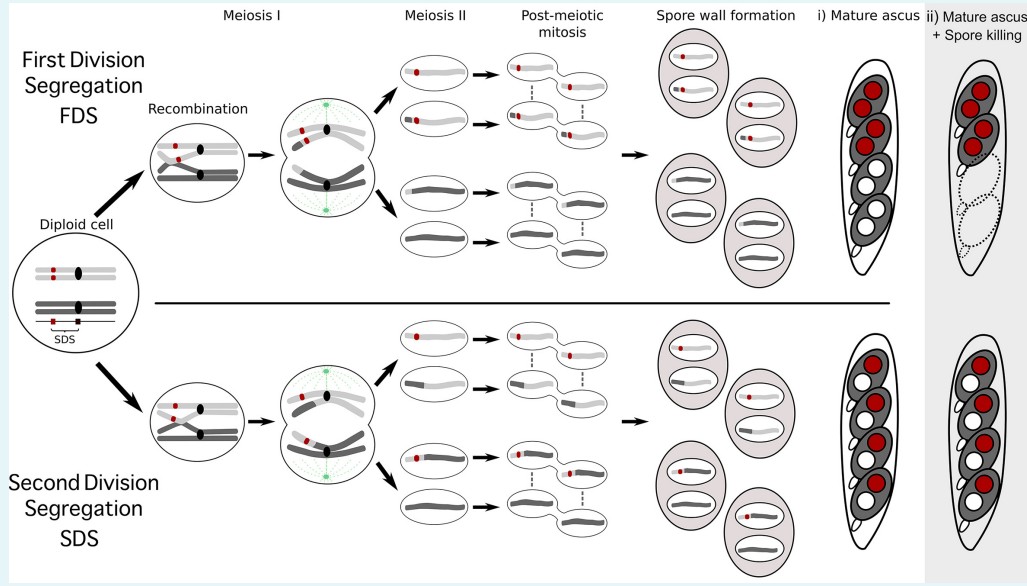

**Box 1—figure 1.** Schematic representation of meiotic segregation in *Podospora anserina*.

DOI: https://doi.org/10.7554/eLife.46454.003

DOI: https://doi.org/10.7554/eLife.46454.004

reads were mapped to Podan2, all samples had a sequencing depth greater than 75x (**Supplementary file 2**). Taken together, our genome assemblies, resulting from both long- and short-read data, are very comprehensive.

## The *Podospora* species are closely related and highly syntenic

A NeighborNet split network of 1000 single-copy orthologs (including introns) showed that the *P. anserina* samples are remarkably similar to each other and distinct from those of both *P. comata* and *P. pauciseta* (**Figure 1A**). Nevertheless, the three taxa are very closely related: the average genic

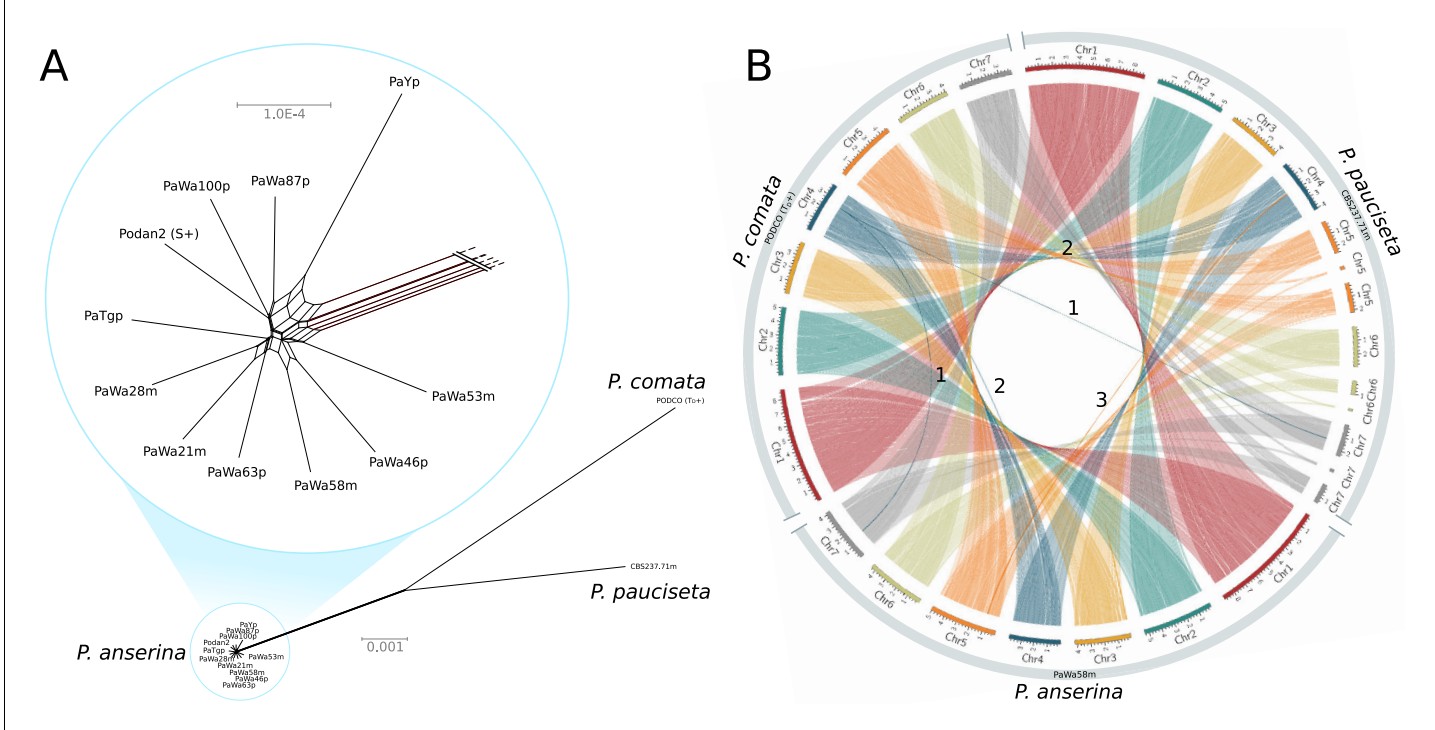

**Figure 1.** Relationship and synteny of *Podospora* strains. (**A**) An unrooted NeighborNet split network based on 1000 orthologous genes of the strains representing the three *Podospora* species shows that the species are distinct from each other but still closely related. A close-up of the cluster of *P. anserina* strains reveals a reticulated relationship and very low genetic diversity (average genic distance of 99.97%). (**B**) A Circos plot of NUCmer alignments (larger than 13 kb) between the reference genome of *P. comata*, the new genome of *P. pauciseta*, and a representative strain of *P. anserina*. Chromosomes 5, 6, and 7 of *P. pauciseta* are not fully assembled, in particular around regions matching the location of the centromere in the *P. anserina* linkage map (not shown). Regardless, the alignment of the assembled region shows highly conserved large-scale synteny between the taxa, with the exception of three large translocation events marked with numbers. Numbers 1 and 2 are potential mis-assemblies in the *P. comata* reference genome, whereas the translocation number 3 between *P. pauciseta* and *P. anserina* corresponds to the *Spok* block. See ***Figure 1—figure supplement 1*** for an equivalent Circos plot of inversions.

DOI: https://doi.org/10.7554/eLife.46454.006

The following figure supplement is available for figure 1:

**Figure supplement 1.** Circos plot showing only inversions (larger than 13 kb) between the reference genome of *P. comata*, the new genome of *P. pauciseta*, and a representative strain of *P. anserina*.

DOI: https://doi.org/10.7554/eLife.46454.007

identity within *P. anserina* is 99.97%, whereas the genic distance between *P. anserina* and *P. pauciseta* is 99.10%, between *P. anserina* and *P. comata* is 98.87%, and between *P. pauciseta* and *P. comata* is 98.79%. Accordingly, the whole-genome alignments recovered strongly conserved synteny at the chromosomal level when small (<13 kb) translocations (presumably due to transposable elements (TEs)) are excluded (***Figure 1B***; ***Figure 1—figure supplement 1***). Only three larger translocations were detected, of which two might be mis-assemblies in the reference genome of *P. comata* (PODCO; see 'Materials and methods').

## Identification and description of *Spok* genes

By searching our assemblies for the *Spok2* sequence (presented by ***Grognet et al., 2014***) using BLAST, we confirm the presence of this *Spok* gene on the left arm of chromosome 5 in the majority of strains, in agreement with ***Grognet et al. (2014)***. Furthermore, on the basis of sequence similarity with *Spok2*, we identified two novel homologs that we refer to as *Spok3* and *Spok4*. These newly identified *Spoks* are found at different genomic locations depending on the strain. Both *Spok3* and *Spok4* can be located on the left arm of chromosome 3 or on the left arm of chromosome 5, and *Spok3* can be found at an additional location on the right arm of chromosome 5. In addition, the

**Table 1.** List of all strains used in this study.

| Sample | Site of origin | Spore killer* | Sequenced | Technology | Mycelium | *Spok* genes† | *Spok* block location‡ | Flanking genes§ |
|---|---|---|---|---|---|---|---|---|
| **Natural isolates¶** | | | | | | | | |
| Wa21– | Wageningen | *Psk-2 (Psk-3)* | DNA | PacBio | Monokaryon | *Spok2, Spok3* | 5R: 3325285 | Pa_5_7950 – Pa_5_7960 |
| | | | | HiSeq 2500 | | | | |
| Wa28– | Wageningen | *Psk-2* | DNA | PacBio | Monokaryon | *Spok2, Spok3* | 5R: 3325285 | Pa_5_7950 – Pa_5_7960 |
| | | | | HiSeq 2500 | | | | |
| Wa46+ | Wageningen | Naïve *(Psk-4)* | DNA | PacBio | Monokaryon | *SpokΨ1* | – | – |
| | | | | HiSeq 2500 | | | | |
| Wa53– | Wageningen | Psk-1 | DNA | PacBio | Monokaryon | *Spok2, Spok3, Spok4* | 3L: 358693 | Pa_3_945 – Pa_3_950 |
| | | | | HiSeq 2500 | | | | |
| Wa58– | Wageningen | Psk-7 | DNA | PacBio | Monokaryon | *Spok2, Spok3, Spok4* | 5L: 896822 | Pa_5_490 – Pa_5_470 |
| | | | | HiSeq 2500 | | | | |
| Wa63+ | Wageningen | *Psk-S* | DNA | PacBio | Monokaryon | *Spok2* | – | – |
| | | | | HiSeq 2500 | | | | |
| Wa63– | Wageningen | *Psk-S* | RNA | HiSeq 2500 | Monokaryon | *Spok2* | – | – |
| Wa87+ | Wageningen | *Psk-1* | DNA | PacBio | Monokaryon | *Spok2, Spok3, Spok4, SpokΨ1* | 3L: 358693 | Pa_3_945 – Pa_3_950 |
| | | | | HiSeq 2500 | | | | |
| Y+ | France | *Psk-5* | DNA | MinION | Monokaryon | *Spok3, Spok4* | 3L: 358693 | Pa_3_945 – Pa_3_950 |
| | | | | HiSeq 2500 | | | | |
| Wa100+ | Wageningen | *Psk-8* | DNA | PacBio | Monokaryon | *Spok2, Spok4, SpokΨ1* | 5L: 896822 | Pa_5_490 – Pa_5_470 |
| | | | | HiSeq 2500 | | | | |
| T$_G$+ | France | *Psk-5 (sk-1)* | DNA | MinION | Monokaryon | *Spok3, Spok3, Spok4* | 3L: 358693 | Pa_3_945 – Pa_3_950 |
| | | | DNA | HiSeq X | | | | |
| CBS237.71– | Israel | *Psk-P1* | DNA | MinION | Monokaryon | *Spok2, Spok3* | 4R: 1674812 | Pa_4_3420 – Pa_4_3410 |
| | | | DNA | HiSeq X | | | | |
| T$_D$+ | ? | *Psk-C1* | DNA | HiSeq X | Monokaryon | *Spok1* | – | – |
| | | | RNA | HiSeq 2500 | | | | |
| S+ | France | *Psk-S* | DNA | HiSeq X | Monokaryon | *Spok2* | – | – |
| S– | France | *Psk-S* | DNA | HiSeq X | Monokaryon | *Spok2* | – | – |
| Wa47 | Wageningen | naïve *(Psk-6)* | – | – | – | Not sequenced | – | – |
| Z | France | *Psk-7* | – | – | – | Not sequenced | – | – |
| s | France | *Psk-S* | – | – | – | Not sequenced | – | – |
| Us5 | Germany | *Psk-S* | – | – | – | Not sequenced | – | – |
| **Backcrosses to S††** | | | | | | | | |
| Psk1xS$_5$- (Wa53) | | Psk-1 | DNA | HiSeq 2500 | Monokaryon | *Spok2, Spok3, Spok4* | 3L: 358693 | Pa_3_945 – Pa_3_950 |
| Psk2xS$_5$+ (Wa28) | | Psk-2 | DNA | HiSeq 2500 | Monokaryon | *Spok2, Spok3* | 5R: 3325285 | Pa_5_7950 – Pa_5_7960 |

*Table 1 continued on next page*

*Table 1 continued*

| Sample | Site of origin | Spore killer* | Sequenced | Technology | Mycelium | Spok genes[†] | Spok block location[‡] | Flanking genes[§] |
|---|---|---|---|---|---|---|---|---|
| Psk5xS$_5$+ (Y) | | Psk-1 (Psk-5) | DNA | HiSeq 2500 | Monokaryon | Spok2, Spok3, Spok4 | 3L: 358693 | Pa_3_945 – Pa_3_950 |
| Psk7xS$_5$+ (Wa58) | | Psk-7 | DNA | HiSeq 2500 | Monokaryon | Spok2, Spok3, Spok4 | 5L: 896822 | Pa_5_490 – Pa_5_470 |
| Psk1xS$_{14}$-vsS | | Psk-1 | RNA | HiSeq 2500 | Selfing dikaryon | Spok2, Spok3, Spok4 | Like parental | Like parental |
| Psk2xS$_{14}$-vsS | | Psk-2 | RNA | HiSeq 2500 | Selfing dikaryon | Spok2, Spok3 | Like parental | Like parental |
| Psk5xS$_{14}$-vsS | | Psk-1 | RNA | HiSeq 2500 | Selfing dikaryon | Spok2, Spok3, Spok4 | Like parental | Like parental |
| Psk7xS$_{14}$-vsS | | Psk-7 | RNA | HiSeq 2500 | Selfing dikaryon | Spok2, Spok3, + Spok4 | Like parental | Like parental |

*The spore killer type of each strain is given as reported by **van der Gaag et al. (2000)** when our phenotyping agrees, and in parenthesis when it does not. The S$_{14}$ strains were phenotyped by us.

[†]The S$_{14}$ *Spoks* were inferred from RNAseq mapping.

[‡]The chromosome number and the arm (R for right, and L for left) describing the position of the *Spok* block are given, along with the coordinates in the Podan2 chromosome.

[§] As the exact insertion point is always intergenic, we also provide the flanking genes. The gene nomenclature follows **Espagne et al. (2008)**, where Pa stands for *Podospora anserina*, the number between underscores is the chromosome and the last number is the gene code.

Note that strain s and strain S are different natural isolates.

[††]Parentheses denote parental spore killer strains.

'Like parental' denotes that the location of the *Spok* block in the S$_{14}$ backcrosses was not inferred from sequencing data, however it should correspond to the location in the S$_5$ backcrosses.

'–', Not applicable.

DOI: https://doi.org/10.7554/eLife.46454.005

BLAST searches recovered a pseudogenized *Spok* gene (*Spok*Ψ*1*) in the subtelomeric region of the right arm of chromosome 5. The *Spok* gene content of the strains investigated in this study is reported in *Table 1*.

A schematic representation of the *Spok* homologs is shown in *Figure 2A*. We considered the *Spok2* sequence of S+, and the *Spok3* and *Spok4* sequences of Wa87+ as reference alleles for each homolog. Overall they show a high degree of conservation, including the 3' and 5' UTRs. A nucleotide alignment of the *Spok* genes' coding sequence revealed 130 variable sites out of 2334 total sites (*Figure 2—figure supplement 1*). A relatively large proportion (67%, 87/130) of those variable sites result in amino acid changes and 74% are unique to one of the *Spok* homologs. *Table 2* displays pairwise comparisons of the amino-acid sequence of the SPOK proteins, revealing a high rate of non-synonymous substitutions, and a relatively high similarity between *Spok1* and *Spok4*. There are six indels among all the *Spok* genes, including one at the 5' end of the ORF that represents a variable-length repeat region, and one at the 3' end of the ORF that is shared by *Spok3* and *Spok4*. The 3' end indel induces a frameshift and changes the position of the stop codon (*Figure 2A*). *Spok*Ψ*1* has a missing 5' end, multiple stop codons, and a discoglosse (Tc1/*mariner*-like) DNA transposon (*Espagne et al., 2008*) inserted in the coding region. Of particular interest, *Spok*Ψ*1* has no deletions relative to the other *Spok* homologs, suggesting that the indels in the functional *Spok* homologs represent derived deletions.

There is little allelic variation within the *Spok* homologs in the Wageningen population and the variants of the four homologs cluster phylogenetically (*Figure 2B and C*). The *Spok2* gene in the Wageningen strains are identical to the two alleles described in *Grognet et al. (2014)*, with the exception of *Spok2* from Wa58– which has a single SNP that results in a D358N substitution. The *Spok2* allele of the French strain A, which shows resistance without killing (as reported by *Grognet et al., 2014*), was not found in any of the genomes investigated in this study. *Spok3* has five allelic variants, and the allelic variation of *Spok4* is reminiscent of *Spok2*, with only Wa100+ and

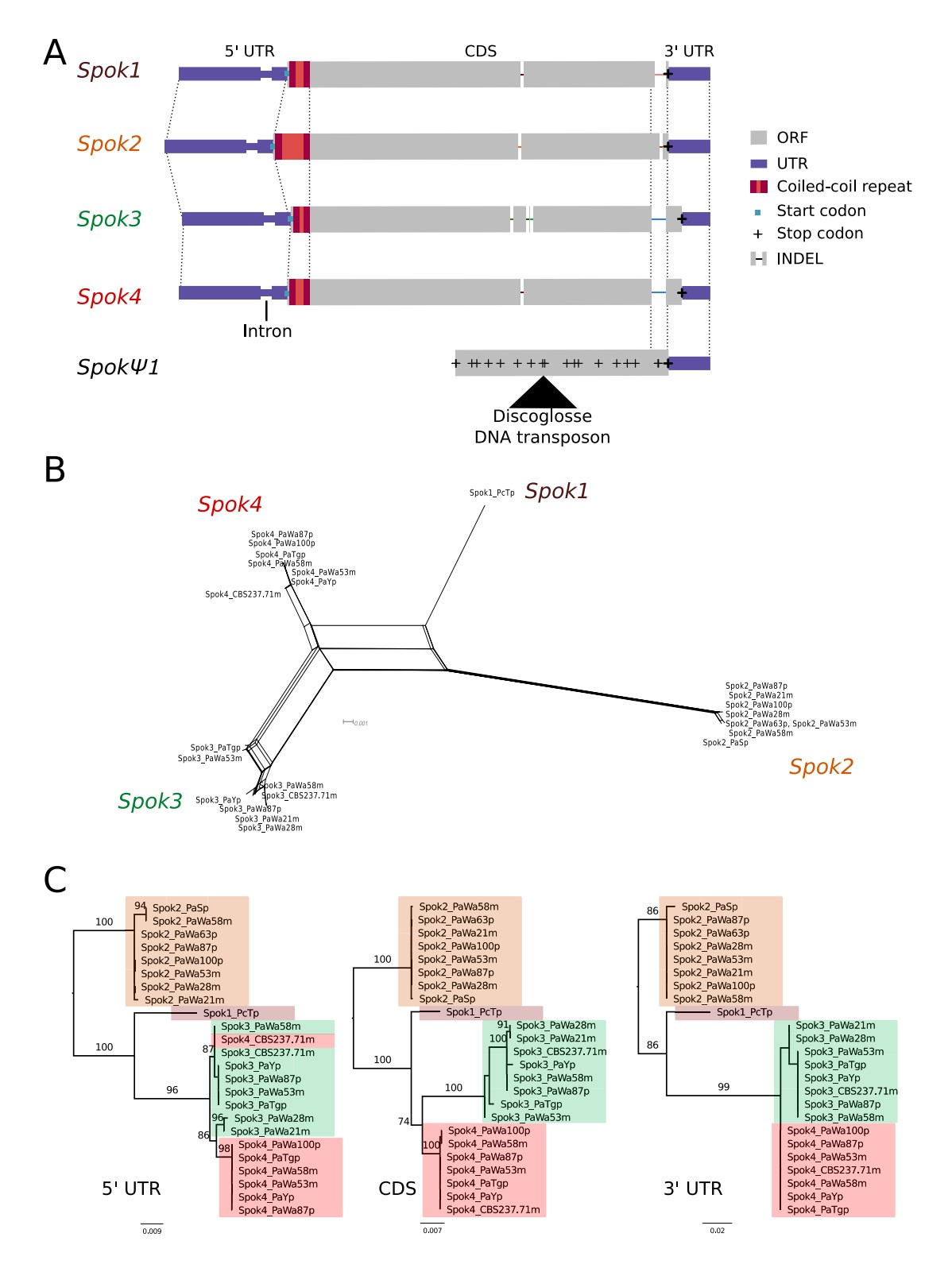

**Figure 2.** Relationships among the *Spok* homologs. (**A**) Schematic representation of the main features of the *Spok* genes. All homologs share an intron within the 5' UTR. At the start of the coding region (CDS) there is a repeat region, in which the number of repeats varies among the homologs. The central portion of the CDS has a number of indels, which appear to be independent deletions in each of *Spok2*, *Spok3*, and *Spok4*. There is a frameshift mutation at the 3' end of the CDS that shifts the stop codon of *Spok3* and *Spok4* into what is the 3' UTR of *Spok1* and *Spok2*. The

*Figure 2 continued*

pseudogenized *Spok* gene (*Spok*Ψ*1*) contains none of the aforementioned central indels and appears to share the stop codon of *Spok1* and *Spok2*. However, there are numerous mutations that result in stop codons within the CDS as well as a full DNA transposon (discoglosse) insertion. No homologous sequence of the 5' end of *Spok*Ψ*1* is present. (**B**) A NeighborNet split network of all active *Spok* genes from all strains sequenced in this study. The four homologs cluster together well, but there are a number of reticulations, which presumably are the result of gene conversion events. (**C**) Maximum likelihood trees based on three separate regions of the *Spok* genes: the 5' UTR, the CDS, and the 3' UTR (starting from the stop codon of *Spok3* and *Spok4*). The trees are rooted arbitrarily using *Spok2*. Branches are drawn proportional to the scale bar (substitutions per site), with bootstrap support values higher than 70 shown above.

DOI: https://doi.org/10.7554/eLife.46454.008

The following figure supplements are available for figure 2:

**Figure supplement 1.** Nucleotide alignment of the *Spok* homologs from the strains sequenced with long-read technologies.

DOI: https://doi.org/10.7554/eLife.46454.009

**Figure supplement 2.** The expression of *Spok* genes based on RNAseq data.

DOI: https://doi.org/10.7554/eLife.46454.010

Wa58– having a single synonymous SNP (*Figure 2C*). Lastly, the three copies of *Spok*Ψ*1* are all unique.

Notably, a number of the allelic variants of *Spok3* show signatures of gene conversion events (*Lazzaro and Clark, 2001*). Specifically, strain Y+ has three SNPs near the start of the gene that result in amino-acid changes and that match those in *Spok2* exactly (*Figure 2—figure supplement 1*). The Wa53+ allele of *Spok3* has a series of SNPs (a track of 205 bp) that are identical to those in *Spok4* but different from all other *Spok3* sequences, and three additional SNPs near the 5' end that also match *Spok4* (*Figure 2—figure supplement 1*). The $T_G$+ strain possesses two identical copies of *Spok3* (see 'Materials and methods') that share the aforementioned tract with Wa53+, but which extends for an additional 217 bp (*Figure 2—figure supplement 1*). These chimeric *Spoks* are recovered from the final assemblies (pre- and post-Pilon polishing) with high long-read coverage (>30 x), suggesting that our finding is not a bioinformatic artifact. The gene conversion events between *Spok* homologs are supported by the reticulation shown in a NeighborNet split network (*Figure 2B*) and by a significant recombination Phi test (199 informative sites, p=1.528e-12). A maximum likelihood phylogenetic analysis of the UTR sequences (defined by conservation across homologs) suggests that *Spok3* and *Spok4* are closely related (*Figure 2C*), which is at odds with the high structural similarity of the coding sequences of *Spok1* and *Spok4* (*Figure 2A*). Therefore, we cannot make any strong inference about the relationships between the *Spok* homologs from the sequence data.

In the few strains with no copy of *Spok2*, analysis of the region suggests that this is a result of a one-time deletion (*Figure 3*). The annotation in the original reference genomes of $T_D$ and S is erroneous because of mis-assemblies and/or incomplete exon prediction, which were both corrected in our study using our own Illumina data and annotation pipeline, and then validated by the RNAseq expression data for $T_D$. First, the flanking gene P_5_20 (marked as (1) in *Figure 3*) in *P. pauciseta* (CBS237.71) and *P. comata* ($T_D$) is considerably longer than the *P. anserina* ortholog, which is truncated by a discoglosse (Tc1/*mariner*-like) DNA transposon (2). In the strains without *Spok2* (Wa46, Y, and $T_G$), the discoglosse transposon itself is interrupted and the sequence continues on the 3' end of a fragmented crapaud (*gypsy*/Ty3) long terminal repeat (LTR) element, which can be found in full length downstream of *Spok2* in the other strains. This configuration implies that the absence of

**Table 2.** Pairwise statistics between SPOK homologs.

The $d_N/d_S$ ratios, averaged across the coding region, are shown below the diagonal; pairwise amino acid changes are shown above.

| | SPOK4 | SPOK3 | SPOK2 | SPOK1 |
|---|---|---|---|---|
| SPOK4 | x | 41 | 53 | 19 |
| SPOK3 | 0.8404081 | x | 54 | 51 |
| SPOK2 | 0.9731409 | 0.9771488 | x | 40 |
| SPOK1 | 0.6593501 | 0.7833958 | 0.7851462 | x |

DOI: https://doi.org/10.7554/eLife.46454.011

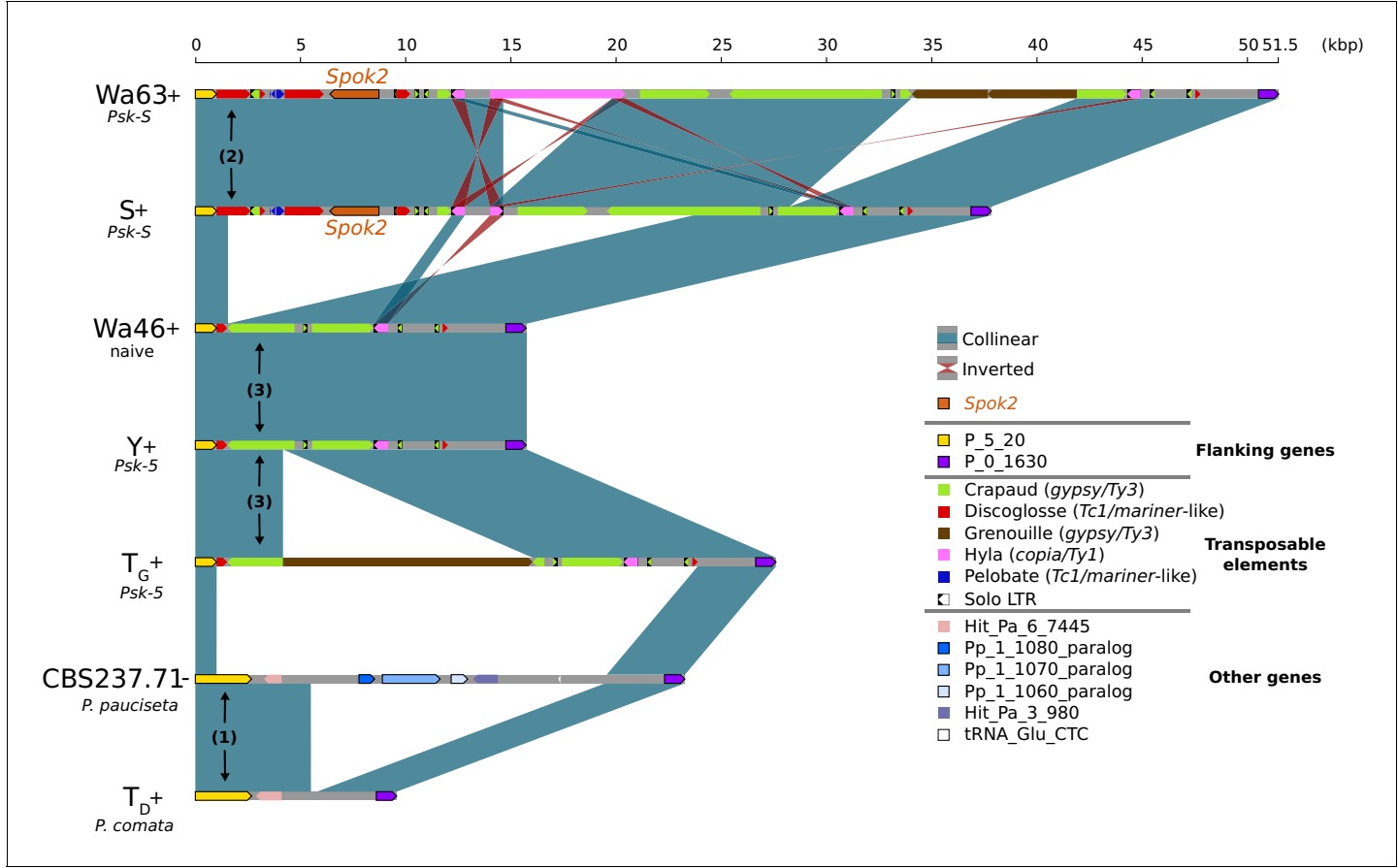

**Figure 3.** Alignment of the *Spok2* locus in selected strains. The plot displays pairwise comparisons of *Spok2* haplotypes with blue segments connecting syntenic regions of homology and red segments showing inversions. The haplotypes are defined by the flanking genes P_5_20 and P_0_1630 located on chromosome 5 of the three sampled species. Every strain has a haplotype of different size, mainly owing to differences in their transposable element (TE) content. Within *P. anserina*, the TE variation across all sequenced strains occurs downstream of *Spok2*, as exemplified by strains Wa63 and S. The strains Wa46, Y and $T_G$ all lack *Spok2* and share break points. See main text for a hypothesis of events (numbered). Notice that P_5_20 stands for the genes Pa_5_20 and PODCO_500020 in the reference annotation of *P. anserina* and *P. comata*, respectively, while P_0_1630 stands for Pa_0_1630 and PODCO_001630. As a note, *P. pauciseta* has a duplication of three genes in tandem from chromosome 1 (Pa_1_1080–60) between the flanking genes. Hit_Pa_X_XXX genes stand for significant BLAST hits to genes of Podan2. TE nomenclature follows *Espagne et al. (2008)*.

DOI: https://doi.org/10.7554/eLife.46454.012

The following source data is available for figure 3:

**Source data 1.** Annotation file for TEs surrounding *Spok2*.

DOI: https://doi.org/10.7554/eLife.46454.013

*Spok2* constitutes a deletion (3), rather than the ancestral state within *P. anserina*. An alternative scenario would require multiple additional insertions and deletions of TEs and *Spok2*.

The *Spok1* gene was previously identified from the *P. comata* strain $T_D$ (*Grognet et al., 2014*). No other strains investigated in this study were found to possess *Spok1*, indicating that this gene is probably not present in *P. anserina*. Remarkably, BLAST searches of the *Spok2* gene (including the UTR sequences) revealed the presence of a small piece (~156-bp long) of a presumably degraded *Spok* gene in the $T_D$ de novo assembly and on chromosome 4 of PODCO. This piece overlaps with the last amino acids of the CDS 3' end and is flanked by an arthroleptis (solo LTR) retrotransposon on one side and by unknown sequence on the other. Owing to the small size of this piece, it not clear whether it belongs to a novel *Spok* gene, but the location (between genes PODCO_401390 and PODCO_401400) differs from those of the other known homologs. The sequencing reveals that the genome of the *P. pauciseta* strain CBS237.71 contains both *Spok3* and *Spok4* (*Table 1*;

*Figure 2B*), but they are at a genomic location that differs from those in any of the *P. anserina* strains.

## *Spok3* and *Spok4* function as meiotic drive genes

We constructed knock-in and knock-out strains to confirm that the newly discovered *Spok* homologs, *Spok3* and *Spok4,* can induce spore killing on their own (*Supplementary file 3*), as previously shown for *Spok2* by *Grognet et al. (2014)*. First, the *Spok2* gene was deleted from the strain s to create a Δ*Spok2* strain for use with the knock-ins. A cross between s and the Δ*Spok2* strain resulted in about 40% two-spored asci, as previously reported by *Grognet et al. (2014)* (80/197, 40.6%) (*Figure 4—figure supplement 2B*). The *Spok3* and *Spok4* genes were inserted separately at the centromere-linked *PaPKS1* locus (a gene controlling the pigmentation of spores [*Coppin and Silar, 2007*]). As *PaPKS1* is tightly linked to the centromere, we expected that if the genes are capable of meiotic drive, then crosses to the Δ*Spok2* strain should yield nearly 100% two-spored asci with white (unpigmented) spores. Accordingly, both *Spok3::PaPKS1* Δ*Spok2* x Δ*Spok2* and *Spok4::PaPKS1* Δ*Spok2* x Δ*Spok2* crosses yielded almost 100% two-spored asci with two white spores (118/119, 99.1%; *Figure 4—figure supplement 2C*) and (343/346, 99.1%; *Figure 4—figure supplement 2D*), respectively. These results show that *Spok3* and *Spok4* function as spore killers when introduced as a single copy at the *PaPKS1* locus .

## The *P. anserina Spok* homologs are functionally independent

To determine whether there are any interactive effects between *Spok2*, *Spok3*, and *Spok4*, we made use of the knock-in strains to assay pairwise interactions among them. First, to determine the interaction between *Spok3* and *Spok4*, we crossed a strain bearing *Spok4* at *PaPKS1* with a strain bearing *Spok3*. Because crosses that are homoallelic for the *PaPKS1* deletion have poor fertility, we constructed a strain in which *Spok3* is inserted as a single copy at the *PaPKS1* locus but just downstream of the coding region (*Spok3::PaPKS1d*) in order to yield strains with normal pigmentation and normal fertility in crosses to *PaPKS1*-deletion strains. In control crosses, the *Spok3::PaPKS1d* strain showed killing when crossed with a strain lacking *Spok3* but no killing when crossed with *Spok3::PaPKS1* (*Figure 4—figure supplement 2E and F*). The cross between *Spok3::PaPKS1d* and *Spok4::PaPKS1* yielded asci that had four aborted spores, indicating mutual killing of *Spok3* and *Spok4* (*Figure 4—figure supplement 2G*). To determine the killing relation between *Spok2* and *Spok3*, a cross was conducted between *Spok3::PaPKS1* and strain s (of the *Psk-S* type). This cross yielded mostly two-spored asci with two unpigmented spores (163/165 asci: 98.8%) (*Figure 4—figure supplement 2H*), indicating that *Spok3* kills in the presence of *Spok2*. Similarly, to determine the killing relation between *Spok2* and *Spok4*, a cross was conducted between *Spok4::PaPKS1* and s, which resulted in 99:5% killing (216/217 asci) (*Figure 4—figure supplement 2I*). Although these two crosses indicate that *Spok2* does not confer resistance to *Spok3* and *Spok4* (*Spok3* and *Spok4* both kill *Spok2*), they do not allow us to determine whether *Spok3* or *Spok4* confer resistance to *Spok2*. To address this point, *Spok2* killing was analyzed in a cross that was homoallelic for *Spok3* (*Spok3::PaPKS1* x *Spok3::PaPKS1d* Δ*Spok2*), which yielded 46% two-spored asci (143/310), confirming that *Spok2* killing occurs in the presence of *Spok3* (*Figure 4—figure supplement 2J*). Finally, to determine whether *Spok4* is resistant to *Spok2*, we made a *Spok4::PaPKS1* x *Spok4::PaPKS1* Δ*Spok2* cross, which resulted in 11/24 two-spored asci (*Figure 4—figure supplement 2K*). Although this genetic background is ill-suited for determining killing frequency (because of the aforementioned effect of the homozygous *PaPKS1* deletion on fertility), the presence of two-spore asci suggests that *Spok4* does not confer resistance to *Spok2* killing. Overall, these results indicate that *Spok2*, *Spok3*, and *Spok4* do not interact.

## The *Spok*s are the Spore-Killer genes of the *psk*s

To evaluate whether the newly discovered *Spok* homologs represent the genes that underlie the *Psk* spore-killer types, we sequenced backcrossed laboratory strains using Illumina Hiseq technology. A strain of each of *Psk-1*, *Psk-2*, *Psk-5* and *Psk-7* was previously backcrossed five times to the reference strain S (*van der Gaag et al., 2000*). The backcrossed strains are referred to here as Psk1xS$_5$, Psk2xS$_5$, Psk5xS$_5$, and Psk7xS$_5$ (*Table 1*). The backcrossed strains should maintain the killing percentage and mutual interactions of the dominant *Psk* parent. Given previous studies, we do not expect S

(*Psk-S*) to be dominant over the other *Psks* (**van der Gaag et al., 2000**). Notably, crossing results reveal that Psk5xS$_5$ has neither a *Psk-5* nor a *Psk-S* phenotype, but a *Psk-1* phenotype (**Figure 4— source data 2**). This is only possible if multiple killing loci are involved, which is concordant with the observation of multiple *Spok* genes in these strains.

Our Illumina data recovered a total of 41,482 filtered biallelic SNPs from the four S$_5$ backcrosses and the parental strains. All backcrossed strains show a few continuous tracts of SNPs from the dominant killer parent (**Figure 4—figure supplement 3**). For example, Psk1xS$_5$– has a long tract in chromosome 1 that represents the *mat–* mating type, which is to be expected because the published reference of S (Podan2), for which the SNPs are called, is of the opposite mating type (*mat+*). Importantly, the location of *Spok3* and/or *Spok4* of each parental strain has a corresponding introgressed SNP tract in the corresponding S$_5$ backcross, while all backcrossed strains possess the *Spok2* gene from strain S (**Figure 4—figure supplement 3**). The *Psk-5* parental strain of Psk5xS$_5$ (strain Y) does not possess *Spok2*, whereas Psk5xS$_5$ does. Hence, the change in the killing phenotype of the backcrossed strain can be attributed to the presence of *Spok2* (see below). Taken together, these data suggest that the total *Spok* gene content is responsible for the killer phenotype of *Psk-1*, *Psk-2*, *Psk-5*, and *Psk-7* (**Figure 4**). In addition, we determined (on the basis of experimental crosses) that the newly described *Psk-8* type can also be described by *Spok* gene content and position (**Figure 4— source data 2** and **3**). Specifically, *Psk-8* has the same *Spok* block position as *Psk-7*, but does not possess *Spok3* (**Figure 4**).

Our results from the crosses also identified inconsistencies with previous studies (see also Appendix 2). Originally, *Psk-4* was defined as a spore-killer (**van der Gaag et al., 2000**. However, the *Psk-4* strain Wa46 has no intact *Spok* genes (**Table 1**). The spore killing observed when this strain was crossed to S in previous publications (or to Wa63 here) is a result of *Spok2*-induced killing. Hence, we recommend discontinuing the use of *Psk-4* and that the term 'naïve' strain is used instead. Moreover, our crossing data show that our representative strain of *Psk-3* (Wa21) (**van der Gaag et al., 2000**) is of *Psk-2* killer type because it does not exhibit spore killing when crossed to Wa28 (*Psk-2*), because it has the expected spore-killing percentage when crossed to a *Psk-S* strain, and because its *Spok* content and location are representative of a *Psk-2* strain. Finally, our representative strain of *Psk-6* (Wa47) behaves as naïve (*Psk-4*), and does not exhibit the spore-killing reported by **van der Gaag et al. (2000)** in test crosses with Wa46 (**Figure 4—source data 2**).

As each isolate of the entire Wageningen collection was previously assessed to determine its *Psk* type (**van der Gaag et al., 2000**), we can estimate the frequency of each *Spok* gene in the Wageningen population. Isolates of *Psk-1*, *Psk-2*, *Psk-4*, *Psk-5*, and *Psk-7*, as well as those previously considered as 'sensitive' (now *Psk-S*), account for 92 of the 99 strains collected from Wageningen. The seven remaining strains were identified as either *Psk-3* or *Psk-6*. Following the rationale outlined in the previous paragraph, we assume that strains annotated as *Psk-4* possess no functional *Spok* genes and omit all the *Psk-3* strains (except Wa21) and the *Psk-6* strains (except Wa47) from the analysis. We estimate that *Spok2* is in 98%, *Spok3* in 17%, and *Spok4* in 11% of the Wageningen strains. A subsample of 11 strains from the 1937 French collection (including strains Y, Z and T$_G$) have also been assessed for their *Psk* type , as have eight strains from a collection from Usingen (Us), Germany (**Hamann and Osiewacz, 2004**; **van der Gaag et al., 2000**). Hence, we infer that *Spok2* is present in all of the Us strains and in 73% of the analyzed French strains. *Spok3* and *Spok4* is in 36% of the French strains, whereas *Spok3* is in one Us strain and *Spok4* is absent from the Us strains.

## *Spok3* and *Spok4* are found in a large region associated with the *Psk* phenotypes: the *Spok* block

Although the *Spok* genes are often assembled into small fragmented contigs when obtained by using Illumina data alone, in the PacBio and MinION assemblies, *Spok3* and *Spok4* are fully recovered within an inserted block of novel sequence (74–167 kbp depending on the strain), hereafter referred to as the *Spok* block (**Figure 4**). When present, the *Spok* block was found once per genome and always contained at least one *Spok* gene. Whole-genome alignments revealed that the *Spok* block has clear boundaries, and is localized at different chromosomal positions on chromosome 3 or on either arm of chromosome 5 in different strains of *P. anserina* (**Table 1**). Importantly, these positions correspond to a single SNP tract identified in the S$_5$ backcrosses. In *P. pauciseta* (CBS237.71), the *Spok* block is found on chromosome 4. This is evident in **Figure 1B** as the only large-scale

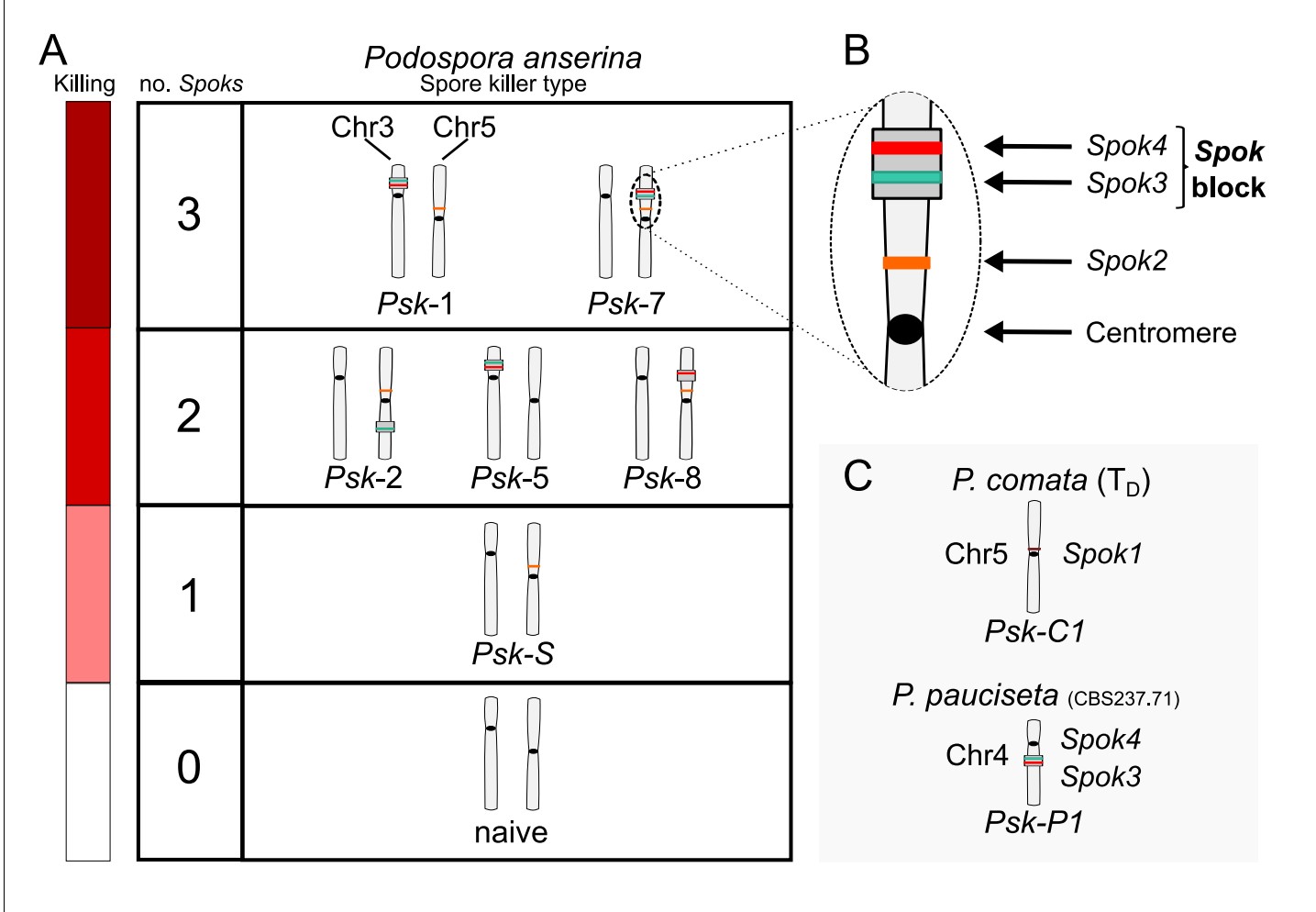

**Figure 4.** Interactions among the various *Psk* types and the occurrence of *Spok* genes. (**A**) The boxes represent hierarchical levels that increase in killing dominance from bottom to top, which correlate with the number of *Spok* genes that a strain possesses. Strains with three *Spok* genes induce the spore-killing of strains with only two *Spok* genes and show mutual resistance to each other. Strains with two *Spok* genes show mutual killing among themselves due to the different *Spok* genes and kill strains with only *Spok2*. Strains with one *Spok* kill strains with no *Spok* genes (naïve strains). The chromosome diagrams depict the presence of the *Spok* genes and their location in the genome for the sequenced strains. (**B**) A zoomed-in look at Chromosome 5 of a *Psk-7* strain, demonstrating that *Spok3* and *Spok4* are present in the *Spok* block and that *Spok2* is present at the standard location. (**C**) The closely related species *P. comata* and *P. pauciseta* also possess *Spok* genes, but at different locations. The *Spok* genes in *P. pauciseta* are present in a smaller *Spok* block, whereas *Spok1* is found on its own and exclusively in *P. comata*.

DOI: https://doi.org/10.7554/eLife.46454.014

The following source data and figure supplements are available for figure 4:

**Source data 1.** Table showing which type of data was used to infer pairwise interactions amongst all *Psks*.

DOI: https://doi.org/10.7554/eLife.46454.020

**Source data 2.** Table with killing percentages for all crosses tested between strains.

DOI: https://doi.org/10.7554/eLife.46454.021

**Source data 3.** Table with observations of killing to determine pairwise interactions of the *Psks*.

DOI: https://doi.org/10.7554/eLife.46454.022

**Figure supplement 1.** Killing hierarchies among *Podospora* spore killers.

DOI: https://doi.org/10.7554/eLife.46454.015

**Figure supplement 2.** Genetic manipulations of *Spok* genes in the s strain background.

DOI: https://doi.org/10.7554/eLife.46454.016

**Figure supplement 3.** Chromosomal segments remaining in the genomes after backcrossing of spore-killer strains into the S background, along with the parental strains.

DOI: https://doi.org/10.7554/eLife.46454.017

**Figure supplement 4.** Crossing design to determine killing interactions between strains.

*Figure 4 continued on next page*

*Figure 4 continued*

DOI: https://doi.org/10.7554/eLife.46454.018

**Figure supplement 5.** Plot comparing pooled sequencing data from the progeny of two-spored asci (left, n = 21) and the progeny of four-spored asci (right, n = 63) from a cross of *Psk-1* (Wa87+) and *Psk-5* (Y–).

DOI: https://doi.org/10.7554/eLife.46454.019

translocation between *P. anserina* and *P. pauciseta* (number 3). The *Spok* blocks of the different strains share segments and overall structure (**Figure 5** and **Figure 5—figure supplement 2**), which suggests that they have a shared ancestry. However, complex rearrangements are found when aligning the block between the genomes. Within the *Spok* block, a given strain can harbor either or both of *Spok3* and *Spok4*, and the regions containing the *Spok* genes appear to represent a duplication event (**Figure 5**). Strain T$_G$+ shows an additional duplication that has resulted in a second copy of *Spok3* (**Figure 5—figure supplement 2**). When present, *SpokΨ1* is surrounded by numerous TEs, and the region does not appear to be homologous to the *Spok* block (**Figure 5—figure supplements 1** and **3**).

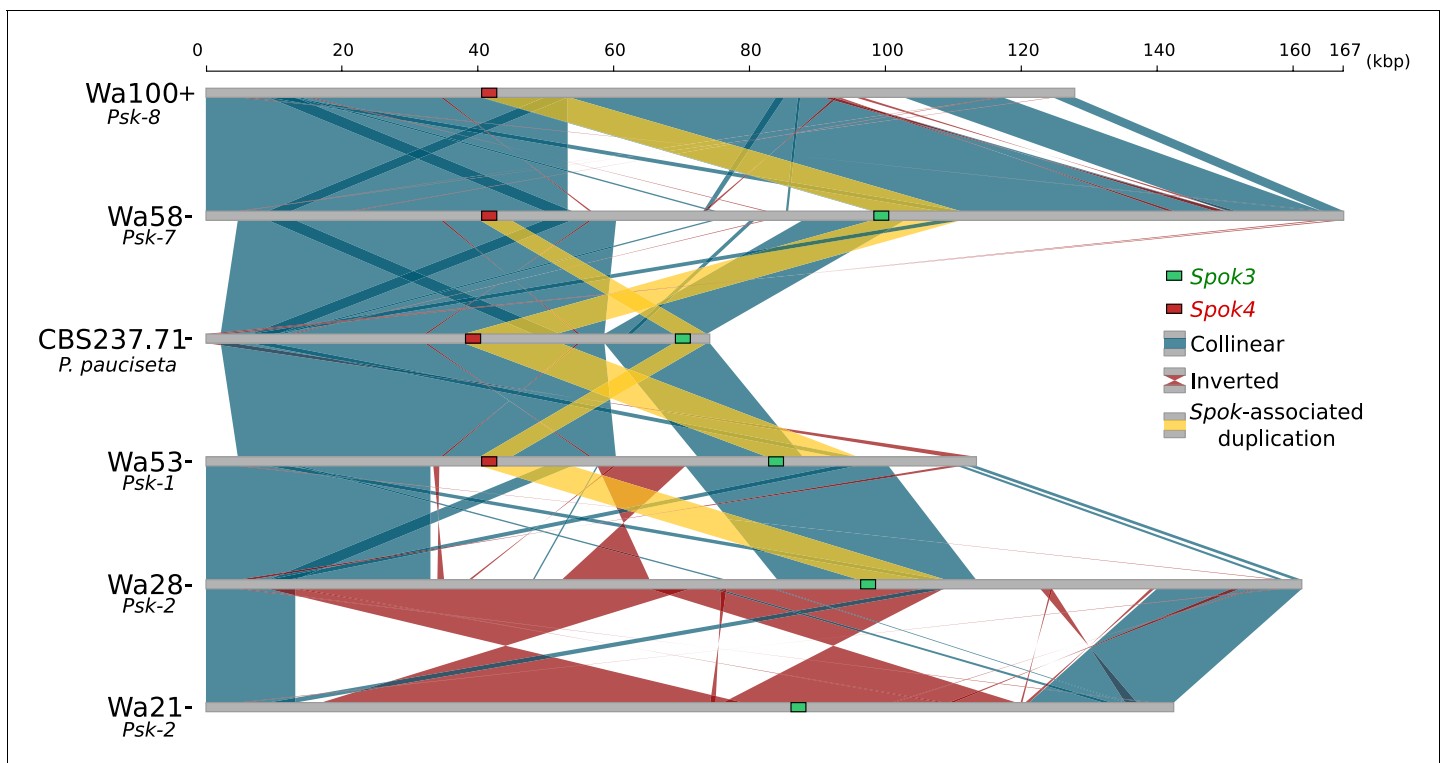

**Figure 5.** Alignment of the *Spok* blocks from different strains. Gray bars represent the block sequences, blue vertical lines connect collinear regions between blocks, while red lines indicate inverted regions. The yellow lines show the region that is duplicated within the block surrounding *Spok3* (green) and *Spok4* (red).

DOI: https://doi.org/10.7554/eLife.46454.023

The following figure supplements are available for figure 5:

**Figure supplement 1.** Alignment of chromosome 5 for three representative *P. anserina* strains (without size filtering).

DOI: https://doi.org/10.7554/eLife.46454.024

**Figure supplement 2.** Alignment of the *Spok* block from the *Psk-1* and *Psk-5* strains shows high overall collinearity.

DOI: https://doi.org/10.7554/eLife.46454.025

**Figure supplement 3.** Dot plot comparing the Wa87– *Spok* block (between and excluding genes Pa_3_945 and Pa_3_950) to the region containing *SpokΨ1* in Wa87– (Pa_5_10570 and Pa_5_10565).

DOI: https://doi.org/10.7554/eLife.46454.026

## All *Psk* interactions can be explained by the presence, absence, or location of *Spok2*, *Spok3*, and *Spok4*

To determine whether other components of the *Spok* block influence the interactions of the *Spok* genes and/or the *Psks*, we conducted crosses between selected strains and evaluated spore killing (*Figure 4—source data 1*, *2* and *3*). Specifically, dikaryotic F1 progeny that are homoallelic for the killing locus were selected, backcrossed to both parental strains, and also allowed to self-fertilize (*Figure 4—figure supplement 4*). On the basis of the results of these crosses, killing interactions were classified into one of the following categories.

1. Mutual killing — spore-killing is observed when F1 strains are backcrossed to either parent, or when F1 progeny are selfed (*Figure 4—figure supplement 4B*).
2. Mutual resistance — no spore-killing is observed when F1 strains are backcrossed to either parent nor when selfed.
3. Dominance interaction — spore-killing is observed when F1 strains are backcrossed to only one of the parental strains, and no spore killing is observed upon selfing.

As an example, in a cross between strains of *Psk-1* and of *Psk-7* killer types, there is spore-killing. However the F1 progeny from this cross show no killing when crossed to either parent, nor when selfed, satisfying condition 2. Thus they are mutually resistant, which is consistent with the fact that they carry the same three *Spok* genes. The reason spore killing is observed in the original cross is because the *Spok* block is located at different genomic positions. As a result, the *Spok* block can co-segregate during meiosis, leaving two spores without any *Spoks* and making them vulnerable to killing (see Appendix 1 for a detailed explanation). An example in which the *Psks* show dominance is in the interaction between *Psk-7* and *Psk-8*, which is revealed by the absence of killing when an F1 progeny from a *Psk-7* x *Psk-8* cross is selfed or crossed to the *Psk-7* parent, and the observation of killing when crossed to the *Psk-8* parent. This result indicates that the F1 progeny inherited its killing function from the *Psk-7* parent, and is consistent with the idea that *Spok* gene interaction determine killing as, although *Psk-8* and *Psk-7* strains have a *Spok* block in the same location, the *Psk-8* version of the block does not possess *Spok3* (*Figure 4*).

The backcrossing method described above was not conducted for representative strains of all pairwise interactions among the *Psks* because of the unmanageable number of crosses that would be involved and the difficulty in mating some strains. In cases where dominance was strongly suspected (on the basis of *Spok* gene distribution, such as with crosses to *Psk-S* or naïve strains), the killing percentage was used as a proxy for dominance because in crosses involving dominance, the killing percentage should reflect that of the dominant *Psk* (*Figure 4—source data 2* and *3*). *Figure 4* displays these results as a killing hierarchy, in which strains of the *Psk* type at the top of the hierarchy are dominant to strains of the *Psk* type lower in the hierarchy and either mutually resistant to (level 3) or mutual killers of (level 2) strains on the same level. The exception is *Psk-5*, which exhibits mutual killing with *Psk-S*. Our classifications, based on crosses, show that the killing hierarchy observed in the Wageningen population of *P. anserina* is an emergent property of the presence and absence of the various *Spok* homologs in the different genomes. Hence, our data demonstrate that other components of the *Spok* block do not affect spore-killing.

Of note, crosses between *Psk-1* and *Psk-5* strains have a lower killing percentage than would be expected on the basis of the FDS of *Spok2* (~25% instead of ~40%). We confirmed that *Spok2* is completely associated with two-spored asci in these crosses using a pooled sequencing approach (*Figure 4—figure supplement 5*), an observation that is in line with the backcrossing results. However, we noted a high prevalence of three-spored asci (which were excluded from the analyses). We also observed three-spored asci in crosses between *Psk-S* and naïve strains. Despite low germination rates, we have been able to isolate a spore from a three-spored ascus in a cross between *Psk-S* and a naïve strain that has no copy of *Spok2* (Appendix 2). Therefore, the three-spored asci are probably due to incomplete penetrance of the killing factor and support the conclusion that the spore killing observed in these crosses is caused by the same gene, *Spok2*. This result is consistent with findings presented in the study by *van der Gaag (2005)* that provided independent evidence for incomplete penetrance of spore-killing between S and Wa46 (*Psk-S* and naïve). The lower killing percentage is probably the result of asci with FDS of *Spok2*, which contain three or four spores instead of two.

### *Spok* interactions among the *Podospora* species

In contrast to the absence of epistatic interactions among the *Spok* genes of *P. anserina*, *Spok1* of *P. comata* and *Spok2* were shown previously to interact epistatically (*Grognet et al., 2014*). To determine whether *Spok1* is also dominant to *Spok3* and *Spok4*, crosses were conducted between strain T$_D$ and strains of *P. anserina*. Although T$_D$ shows low fertility with *P. anserina* (*Boucher et al., 2017*), we were successful in mating T$_D$ to a number of *P. anserina* strains of different *Psk* spore-killer types (*Figure 4—source data 2* and *3*). Often, only a few perithecia were produced with limited numbers of asci available to count, but despite this obstacle, the crosses clearly demonstrate that T$_D$ is dominant over *Psk-S* and *Psk-2* strains, and is mutually resistant to a *Psk-5* strain. These results imply that *Spok1* provides resistance to all of the *Spok* homologs in *P. anserina* and is capable of killing in the presence of *Spok2* and *Spok3*, but not *Spok4*. The mutual resistance with the *Psk-5* strain also demonstrates that *Spok4* provides resistance against *Spok1*. Additional crosses were also conducted with the *P. pauciseta* strain CBS237.71, which were consistent for the *Spok3* and *Spok4* interactions (*Figure 4—source data 2* and *3*). As both T$_D$ and CBS237.71 have unique spore-killing phenotypes, we assign them the types *Psk-C1* and *Psk-P1*, respectively.

### An intron in the 5′ UTR is not required for spore killing

To investigate whether the *Spok* genes are expressed during spore-killing, we conducted an additional nine backcrosses of the S$_5$ strains to S, in order to generate S$_{14}$ backcrossed strains (see 'Materials and methods'). We produced RNAseq data for self-killing S$_{14}$ cultures and mapped the reads to the final assemblies of the dominant killer parental strains (*Figure 2—figure supplement 2A*). The expression of the *Spok* genes is evident in this data and supports the presence of an intron in the 5′ UTR of the *Spok* homologs (*Figure 2* and *Figure 2—figure supplement 2B–E*). Given its conservation across the *Spok* homologs and as the *wtf* spore-killer system in *S. pombe* was described as involving two alternate transcripts of the same gene (*Nuckolls et al., 2017*; *Hu et al., 2017*), the role of the intron in *Spok3* spore-killing activity was investigated. The intron was deleted in a plasmid bearing the *Spok3::PaPKS1* deletion cassette by site-directed mutagenesis, and the modified plasmid was used to transform the ΔKu70 Δ*Spok2* strain. Three transformants bearing the *Spok3* lacking the intron sequence (*Spok3 Δi*) were crossed to a Δ*Spok2* strain. As in the control cross with wildtype (*wt*) *Spok3*, in which close to 100% killing was found, we observed that 109/109 of the asci contained two unpigmented spores (*Figure 4—figure supplement 2L*). Thus, *Spok3 Δi* displays wildtype killing activity. We conclude from this experiment that the unspliced form of *Spok3* is not required for normal killing activity, and neither does the killing and resistance function rely on an alternatively spliced form of this intron.

### Functional annotation of SPOK3 predicts three ordered domains

In order to gain insights into the molecular function of the SPOK proteins, domain identification was performed with HHPred and a HMM profile based on an alignment of 282 *Spok3* homologs from various Ascomycota species. The SPOK3 protein was predicted to be composed of three folded domains (located at amino-acid positions ~ 40 to 170, 210 to 400, and 490 to 700 in the protein) separated by two unstructured domains (~170 to 210 and 400 to 490) as shown in *Figure 6*. No functional identification was recovered for domain 1, but a coiled-coil motif was found in the N-terminal 40 amino acids and predicted to form a parallel dimer, which corresponds to the variable length repeat of the nucleotide sequences (*Figure 2A*). Domain 2 showed homology to a class of phospho-diesterases of the PD-(D/E)XK superfamily (~214 to 325) with the catalytic residues forming the PD-(D/E)XK motif spanning positions 219 to 240 in the SPOK3 sequence (*Steczkiewicz et al., 2012*). The best hit in HHPred was to the HsdR subunit of a type-I restriction enzyme from *Vibrio vulnificus* (*Uyen et al., 2009*). The sequences align in the catalytic core region in the PD-(D/E)XK motif and also around a QxxxY motif (294 to 298 in SPOK3) that was found to be important for nucleic-acid binding and nuclease activity (*Sisáková et al., 2008*) (*Figure 6—figure supplement 2*).

Domain 3 was identified as a hypothetical kinase domain (~539 to 700) as predicted previously by *Grognet et al. (2014)*. In addition, a motif with a cluster of three highly conserved cysteine residues together with histidine residues (C-x3-C-x13-C-x5-H-x7-H) that is reminiscent of the zinc-finger motifs was identified upstream of the kinase motif (*Figure 6*). As previously reported for *Spok2*, D667 was identified as the catalytic base residue in the catalytic loop (subdomain VIb) of the kinase

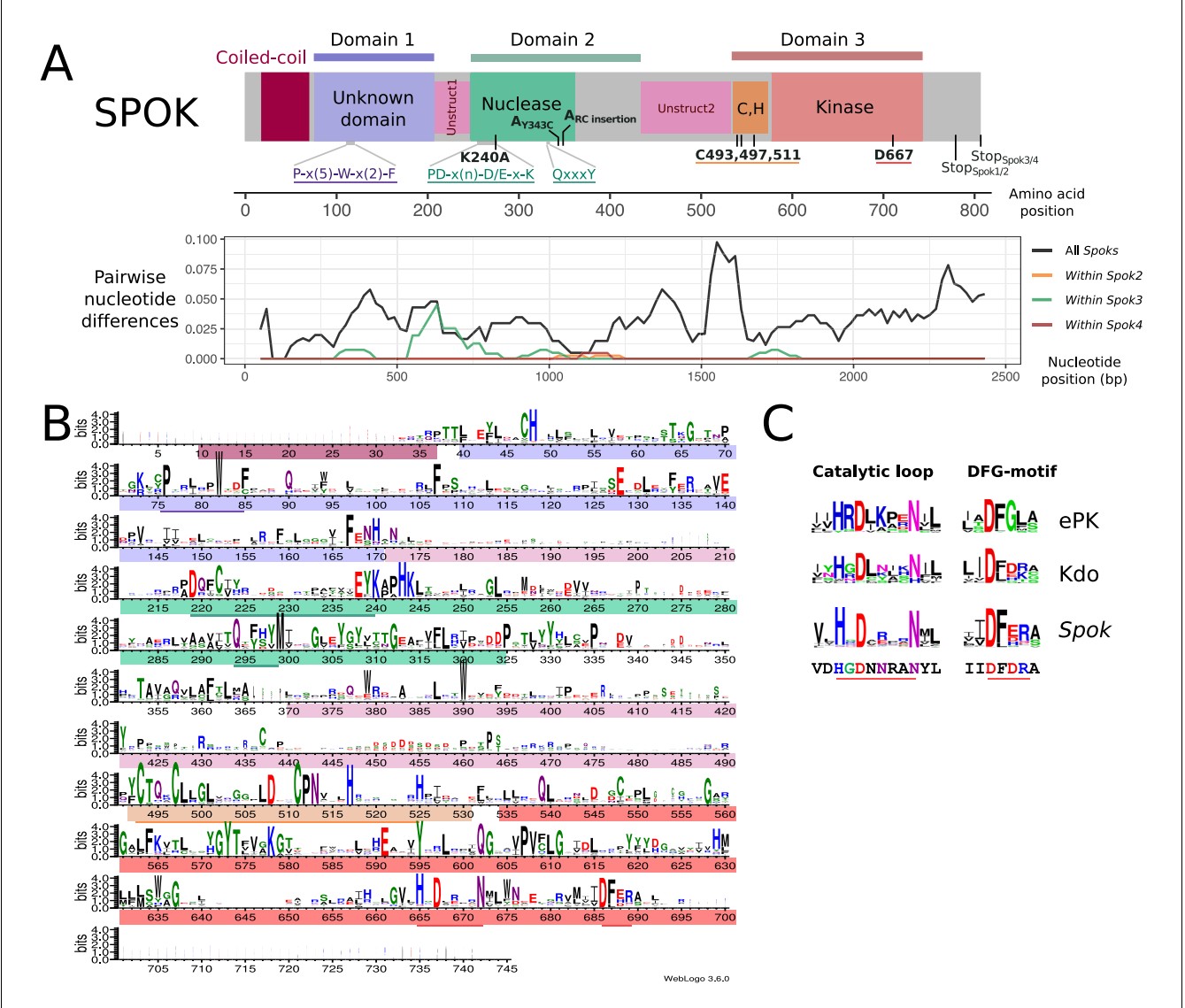

**Figure 6.** Functional annotation of the SPOK protein. (**A**) (Top) A predicted domain diagram of the SPOK protein displays the N-terminal coiled-coil region (in magenta), the N-terminal domain of unknown function (in lilac), the two unstructured regions (in pink), the PD-(D/E)XK nuclease domain (in green), the cysteine cluster region (in orange) and the kinase domain (in red) with coordinates based on the alignment of all SPOK homologs. The positions of key residues and conserved motifs are indicated with the same color code. The mutations labels that are marked in bold correspond to the SPOK3 coordinates, except for the mutations on the SPOK2 of strain A. (Bottom) A plot of the pairwise nucleotide distances between all alleles of a given *Spok* indicates which regions of the protein are conserved or divergent, and where the polymorphisms within a single *Spok* gene are located. The predicted unstructured regions generally show greater divergence. (**B**) HMM profile derived from an alignment of 282 SPOK3 homologs from Ascomycota showing conserved residues. The domains identified in (**A**) are shown with the same color code, and key motifs and residues are underlined. The profile was generated with Web logo v3. (**C**) Comparison of the HHM profiles in the catalytic loop and DFG-motif region in eukaryotic protein kinases and Kdo kinase (an ELK) with the same region in *Spok* homologs. The sequence below corresponds to the SPOK3 sequence.

DOI: https://doi.org/10.7554/eLife.46454.027

The following source data and figure supplements are available for figure 6:

**Source data 1.** Transformation efficiency of *Spok3* manipulations.
DOI: https://doi.org/10.7554/eLife.46454.030
**Figure supplement 1.** Amino-acid alignment of the SPOK proteins from strains sequenced with long-read technologies.
DOI: https://doi.org/10.7554/eLife.46454.028
**Figure supplement 2.** Predicted nuclease domain of SPOK proteins.
DOI: https://doi.org/10.7554/eLife.46454.029

domain. Kinases often use other proteins as substrates, but they may also target small molecules (*Smith and King, 1995*). Inspection of the VIb and VII functional regions, which are informative as regards to kinase substrate specificity, suggests that the *Spok*-kinase domain might be more closely related to eukaryotic-like kinases (ELKs) than to eukaryotic protein kinases (ePKs), raising the possibility that this kinase domain is not necessarily a protein kinase domain and could phosphorylate other substrates (*Steczkiewicz et al., 2012*; *Kannan et al., 2007*).

## The killing and resistance functions can be attributed to separate domains

The ability of the *Spok* genes to perform both killer and resistance functions with a single protein is unique among meiotic drive systems (*Bravo Núñez et al., 2018b*). To investigate the role that the domains 1–3 may play in these two functions, we constructed a number of point mutations and truncation variants of *Spok3* and assayed their ability to kill or provide resistance in vegetative cells. We were able to determine that domain 2 is important for killing activity whereas domain 3 is important for resistance activity.

It had been shown previously that the predicted kinase domain of SPOK2 (*Figure 6*) is involved in the resistance function (*Grognet et al., 2014*). We introduced a point mutation in a plasmid-cloned *Spok3* gene that led to the replacement of the predicted catalytic aspartic acid residue of *Spok3* by an alanine (D667A). The mutant allele was first used to transform a Δ*Spok2* recipient strain. This *Spok3* D667A mutant allele leads to a drastic reduction in transformation efficiency (*Figure 6—source data 1*), whereas the *Spok3* wildtypeallele only moderately affects the number of transformants. As this approach results in random integration and potential multicopy insertion, we also attempted to introduce the mutant *Spok3* D667A allele as a single copy at the *PaPKS1* locus, as described above for wildtype *Spok3*. The initial transformants were heterokaryotic and displayed sectors of abnormal growth that corresponded to unpigmented mycelium, presumably containing nuclei with *Spok3* D667A that inserted at *PaPKS1*. Monokaryotic transformants could be recovered and were tested for killing activity in a cross to a Δ*Spok2*. Four-spored asci with two white and two black spores were observed, suggesting that the D667A mutation abolishes spore killing. However, when the integrated *Spok3* allele was amplified by PCR and sequenced, it appeared that the allele presents a GAG to TAG mutation, leading to a premature stop codon in position 282 (E282stop). This result is consistent with the observation that *Spok3* D667A affects transformation efficiency and is toxic. Moreover, we detected expression of *Spok2* and *Spok1* in the vegetative cells of monokaryotic (self-sterile) cultures, suggesting that *Spok* activity is not restricted to the sexual cycle (*Figure 2—figure supplement 2C and D*). No further attempts to insert the mutant allele at *PaPKS1* were made.

If the toxicity of the *Spok3* D667A allele in vegetative cells is mechanistically related to spore-killing, it is expected that this toxicity should be suppressed by wildtype *Spok3*. Therefore, we assessed whether *Spok3* D667A toxicity in vegetative cells is suppressed by co-expression with wildtype *Spok3*. Co-transformation experiments were set up with *Spok3* D667A used as the transformation vector in the presence or absence of *wt Spok3*. As in the previous experiment, *Spok3* D667A alone was found to affect transformation efficiency, but this effect was suppressed in co-transformations with *Spok3* (*Figure 6—source data 1*). This experiment confirms that *Spok3* D667A is only toxic in the absence of *Spok3*. Therefore, the *Spok*-related killing and resistance activities can be recapitulated in vegetative cells.

We also analyzed the role of the conserved cysteine cluster just upstream of the predicted kinase domain. Three plasmids with point mutations in that region were constructed (a C493A C497A double mutant, and C511A and C511S point mutants), and the mutant alleles were used in transformation assays as previously described for *Spok3* D667A. All three mutants reduced transformation efficiencies as compared to the controls, and this effect was suppressed in co-transformations with *wt Spok3* (*Figure 6—source data 1*). These results suggest that the kinase domain and the cysteine-cluster region are both required for the *Spok*-related resistance function but not for the killing activity. To test this, we constructed a truncated allele of *Spok3* that lacks these two regions: *Spok3*(1–490) (see *Figure 6—figure supplement 1*). The *Spok3*(1–490) allele drastically reduced transformation efficiencies and this effect was suppressed in co-transformations with wildtype *Spok3* (*Figure 6—source data 1*). If, as proposed here, the toxicity and suppression activities assayed in vegetative cells are mechanistically related to spore-killing, then domain 3 appears to be required for the

resistance function but dispensable for the killing activity, which can be carried out by the N-terminal region of the SPOK3 protein (domains 1 and 2).

Next we analyzed the role of the predicted nuclease domain (domain 2) in spore-killing activity. We generated a plasmid with a point mutant that affects the predicted catalytic core lysine residue (K240A). Introduction of this point mutation in the *Spok3*(1–490) allele abolished its killing activity in transformation assays (*Figure 6—source data 1*), suggesting that the predicted nuclease domain is required for killing activity. The *Spok3* K240A mutant was then inserted at the *PaPKS1* locus and the resulting knock-in strain was crossed with a Δ*Spok2* strain (in order to assay killing) and to a *Spok3:: PaPKS1d* strain (to assay resistance) (*Figure 4—figure supplement 2M and N*). In the cross to Δ*Spok2*, no killing was observed: the majority of the asci were four-spored with two white and two black spores (308/379, 81.2%), indicating that the K240A mutation abolishes the spore-killing activity of *Spok3*. In the *Spok3* K240D::*PaPKS1* x *Spok3*::*PaPKS1d* cross, no killing was observed: the majority of the asci were four-spored with two white and two black spores (268/308, 87%). These crosses indicate that the *Spok3* K240A allele has no killing ability but has retained resistance. *Grognet et al. (2014)* reported that strain A bears a mutant allele of *Spok2* that has affected killing ability but retains resistance. The mutations in that allele fall within a conserved region of the predicted nuclease domain (*Figure 6*) and map on predicted structural models in close vicinity to the catalytic lysine residue (K240 in SPOK3) and the other catalytic residues (*Figure 6—figure supplement 2*). Hence, the properties of the *Spok2* allele of strain A provide independent evidence that the nuclease domain of SPOK proteins is involved in killing activity but dispensable for resistance.

## Phylogenetic distribution of *Spok* genes

A BLAST search for closely related homologs of the *Spok* genes across fungi revealed an uneven distribution of related proteins among taxa. Among the available genomes from the Sordariales, we did not find any protein-coding sequences in addition to our newly described *Spok* genes of *Podospora*. By contrast, we found protein-coding sequences with high similarity across other orders of the Sordariomycetes, namely the Xylariales, Glomerellales and Hypocreales, as well as in one species of the Eurotiomycetes, *Polytolypa hystricis* (Poh; *Figure 7*).

We used maximum likelihood analyses to construct phylogenies of the SPOK sequences and of an orthologous gene set of the strains for which we retrieved hits in the BLAST search (*Figure 7*). These phylogenies reveal two notable patterns. First, the SPOK phylogeny shows a high degree of incongruence with the species phylogeny. Moreover, the SPOK phylogeny can be robustly divided into two clades: Clade I and II. Clade I contains the Fs_82228 sequence from *Fusarium solani* (old name *Nectria haematococca*). This sequence was previously introduced into *P. anserina*, and the genetically modified strain produced empty asci when mated to a naïve strain, suggesting that it has a killing action (*Grognet et al., 2014*). Clade II contains the *Podospora* Spok homologs. The sequences from the two clades are disparately distributed in the species phylogeny. The second notable pattern is the distribution of the SPOK sequences within the genomes. The sequences in Clade I are present in single copies in each strain, except for *Fusarium oxysporum* f. sp. *pisi* (Fop), suggesting that they are all orthologs. By contrast, many of the sequences in Clade II are present in multiple copies in each genome. It is particularly interesting to note how many *Spok* homologs from Clade II are present across strains of *F. oxysporum* (Fo) and the number of copies that are found in each genome. Several of the duplicate *Spok* homologs are present on the lineage-specific chromosomes of *Fusarium* that are often associated with pathogenicity (*Armitage et al., 2018*). *Beauveria bassiana* (Bb) also shows a high degree of variability in homolog content among the four strains that have homologs, indicating that the homologs are polymorphic in this species. The insect pathogen *Metarhizium rileyi* (Mr) shows an interesting pattern in that it possesses four divergent homologs, which is in stark contrast to many of the other species(including *Podospora*) that have multiple, though nearly identical, copies. The Clade II *Spok* homologs also appear to diversify within each strain or species in much the same way as the *Spok* genes do in *Podospora*, with variable lengths of the coil-coil repeat region and frameshift mutations near the 3′ end that relocate the stop codon. A few of the sequences may also represent pseudogenes, as evidenced by premature stop codons and/or frameshifts, although these features might also be the result of unidentified introns (*Figure 7*).

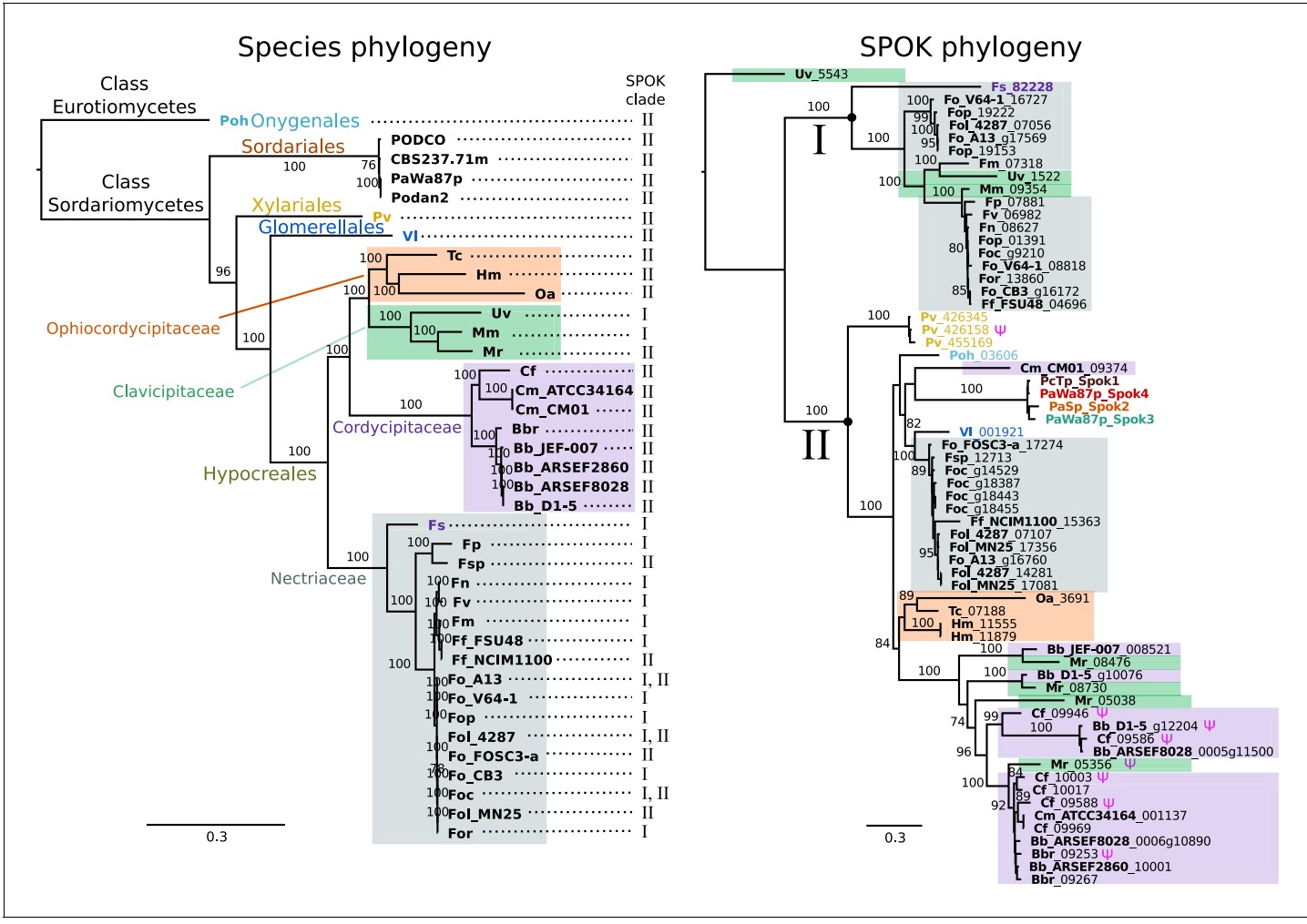

**Figure 7.** The phylogeny of *Podospora* SPOKs and closely related homologs do not follow the species tree. (Left) A maximum likelihood phylogeny of the fungal isolates that harbor *Spok* homologs that are closely related to those of *Podospora* recovered the groupings that are expected on the basis of fungal taxonomic classification (colored boxes and tip labels). The tree was produced using the aligned protein sequence of 288 single-copy orthologs. (Right) A maximum likelihood phylogeny of the SPOK proteins themselves with colors matching the taxonomy in the tree to the left. Two main clades can be distinguished (I and II), and their presence is mapped to each genome in the species phylogeny. Putative pseudogenes are marked with a Ψ symbol. The Fs_82228 protein (in dark purple text) has been demonstrated to exhibit some spore-killing characteristics in a *P. anserina* strain. Rooting of the species tree was based on the split between Classes, whereas the SPOK phylogeny was rooted on the basis of the broader alignment generated for the protein-domain predictions. Bootstrap support values higher than 70 are shown above branches, which are proportional to the scale bar (substitutions per site). SPOK tip labels follow the convention of fungal isolate code (bold) and locus name (see *Figure 7—source data 1* for full species names and genomes).

DOI: https://doi.org/10.7554/eLife.46454.031

The following source data is available for figure 7:

**Source data 1.** Table showing species names and genome labels for codes used in *Figure 7*.

DOI: https://doi.org/10.7554/eLife.46454.032

## Discussion

The identification of *Spok3* and *Spok4* has allowed us to explain the genomic basis for five of the seven *Psk* spore-killer types found in natural populations of *P. anserina*. Through our integrative approach of genomics, molecular biology and phenotyping, we have been able to demonstrate that the multiple drive elements that have been genetically identified in *P. anserina* are not based on different underlying molecular mechanisms and/or specific gene interactions, but rather involve combinations of closely related driver genes that belong to the same *Spok* gene family. The *Spok* genes

thus appear to be responsible for all of the identified drive elements in *Podospora*, with the exception of the *het-s* spore-killing system.

## The *Spok* block

The presence of the complex *Spok* block presents a unique feature among the known meiotic drive systems. Often, meiotic drive elements occupy regions of suppressed recombination that span large tracts of chromosomes (*Turner and Perkins, 1979*; *Hammer et al., 1989*; *Sandler et al., 1959*) and co-occur with complex rearrangements (*Harvey et al., 2014*; *Silver, 1993*; *Dyer et al., 2007*; *Svedberg et al., 2018*). In these well-studied cases, the elements of the drive mechanisms are encoded by separate genes within the region, and the rearrangements and suppression of recombination are expected to have evolved to ensure that the drive machinery (e.g., the toxin and antitoxin genes) is inherited as one unit (*Lyttle, 1991*; *Bravo Núñez et al., 2018b*). In *Podospora*, a single *Spok* gene is fully capable of driving, and thus no region of suppressed recombination is required. Nevertheless, *Spok3* and *Spok4* are found in a large region that is not syntenic with the null allele. Hence, had the *Spok* genes not been previously identified from more placid genomic regions, the entire *Spok* block may have been misidentified as a driving haplotype with multiple interacting components. Considering that single-gene meiotic drivers might be more common than anticipated, it becomes necessary to question whether other drive systems that are located within complex regions, and for which the genetics are not well known, may also represent single gene drivers.

At this stage, our data strongly suggest that the *Spok* block is moving in the genomes as a unit, but nevertheless, the mechanism of movement remains unknown. It may be hypothesized that movement of the block is achieved via an interaction with TEs at different genomic locations and non-allelic homologous recombination. This hypothesis is supported by the observation that the *Spok* genes outside of the *Spok* block, including *SpokΨ1*, are not located at the same position in different species, and that they are often surrounded by similar TEs. Such movement may be under selection as matings between strains that have the same *Spok* genes but in different locations will result in spore-killing. Furthermore, because of the idiosyncrasies of meiosis in *Podospora*, the position of the block may be under selection because the killing frequency is dependent on the frequency of crossing over with the centromere. Alternatively, the TEs may simply accumulate around the *Spok* genes because of a reduced efficacy of purifying selection at regions linked to the driver genes; their presence per se might increase the chance of rearrangements. As such, the role that TEs play in generating complex regions that are associated with meiotic drive should be investigated further in order to determine their importance in the evolution of drive.

## Molecular function of the *Spok* genes

Spore-killing systems display analogies to toxin-antitoxin (TA) systems in bacteria and it is interesting to note that many toxin families rely on nuclease activity (*Harms et al., 2018*). The contrast between the *Spok* system and TA systems, however, resides in the fact that *Spok* toxin and antitoxin activities appear to be supported by the same protein molecule. The predicted kinase activity seems able to counter the toxic activity of the predicted nuclease domain both in cis and in trans. Mechanistic spore-killing models have to explain: (i) how the *Spok* gene can affect spores that do not carry it, at a distance, with SPOK proteins being intracellular proteins (*Grognet et al., 2014*); and (ii) how asymmetry is brought about in this system if killing and resistance activities are carried by the same protein molecule. It is premature to devise a mechanistic model for the molecular basis of *Spok* gene drive, yet it might be possible to conjecture about the substrates of the proposed kinase (and nuclease) activities of SPOK proteins.

A first way to explain how *Spok* genes might act at distance and affect spores not containing them is to hypothesize that toxicity relies on the production of a diffusible metabolite. The predicted nuclease/phosphodiesterase activity would lead to the production of a diffusible toxic molecule that could be further detoxified by phosphorylation (much in the way that bacterial phosphotransferases detoxify antibiotics) (*Shi et al., 2013*). Isolation of the toxic intermediate from the detoxifying activity of the kinase would allow its accumulation specifically outside of the spores expressing SPOK proteins, and thus could bring about asymmetry in spore-killing. Then, alternatively, spores lacking the *Spok* gene might be affected by SPOK proteins if some amount of SPOK protein expressed in the zygote or from sister nuclei is carried over at the time of spore delimitation. The resistance function

of the predicted kinase domain could be explained by hypothesizing that the nuclease activity can be inhibited by autophosphorylation of the SPOK proteins. Alternatively, it could also be that it is the phosphorylation of a distinct macromolecule that nullifies toxicity. In a simple model, the same molecule could be the target of both the kinase and nuclease activity, and the phosphorylation of the target would make it resistant to the toxic action of the predicted nuclease domain. As stated above, models have to explain how killing could occur specifically in spores lacking the *Spok* gene. This situation could occur if the proposed kinase activity is concentration-dependent and favored at higher SPOK protein concentrations (for instance, the kinase activity might require a protein dimerization step that occurs specifically in spores expressing the *Spok* gene). In addition to the yet unresolved mechanistic basis of killing and resistance, the characterization of *Spok* gene function described here poses another puzzle. As all SPOK products have a predicted active kinase, it is not yet known what changes in sequence confer the hierarchical interactions among some *Spok* genes, or why not all SPOKs are able to provide resistance to one another. One possibility is that the cellular targets for the proposed nuclease and kinase activities differ for the different SPOK proteins.

Studies of similar protein domains suggest that the coil-coiled domain is likely to be involved in protein–protein interactions (*van Maldegem et al., 2015*). The fact that *Spok1* and *Spok4* have the same length repeat in this domain could imply that the protein–protein interactions of this domain are important for resistance, as *Spok1* and *Spok4* are mutually resistant. This model would agree somewhat with the results of reporter constructs from *Grognet et al. (2014)*, which showed an N-terminal mCherry tag on *Spok2*-produced empty asci. It is possible that the functional divergence observed between the SPOK proteins is due to mutations in this portion of the protein. In this model, domain 1 might be responsible for the target specificity of the nuclease (and kinase) activity. The killing action itself is expected to be universal among the *Spoks* and is supported by the fact that this entire domain of *Spok3* from $T_G$ is identical to *Spok4*, yet appears to retain *Spok3* functionality. The identification of the role of the predicted nuclease domain in killing and of the predicted kinase domain in resistance provides a first mechanistic insight into the dual role of *Spok* genes. However, further dissection of the molecular action of these proteins is required so that we can fully understand the molecular basis of *Spok* drive.

## Absence of resistance

One of the main factors that stands out in the *Podospora* system, as compared to the other well-studied spore killers, is the lack of resistant strains. Only one strain of *P. anserina* (the French strain A) has ever been described as resistant (*Grognet et al., 2014*). The point mutations of *Spok3* that were induced in the laboratory imply that the creation of a resistant strain is a simple task, as only a single nucleotide change was required. Likewise, the resistant strain A *Spok2* is different from the reference allele by only two novel insertions. Consequently, the lack of resistance does not appear to be the result of a mechanistic constraint. Potentially, the current *Spok* gene distribution could be a relatively young phenomenon and resistance could evolve over time. Another possibility is that resistance itself is somehow costly to the organism and selected against. In addition, it is puzzling that none of the *Spoks* in *P. anserina* show cross resistance. Intuitively, it would seem advantageous for novel *Spok* homologs to evolve new killing functions while maintaining resistance to the other *Spok* homologs. Again, the lack of cross-resistance does not solely appear to be the result of functional constraints, as *Spok1*, which is highly similar to *Spok4*, is resistant to all other *Spok* homologs. It is possible that it is more advantageous to combine multiple independent spore killers than to have a single broadly resistant gene. This option is supported by two observations presented in this study: the occurrence of the killing hierarchy and the association of *Spok3* and *Spok4*. The fact that *Spok3* and *Spok4* are present in the *Spok* block means that they are in tight linkage with each other. It may be the case that the linkage was selected for because it provided strains with the ability to drive against strains with just *Spok3* or just *Spok4*. However, this association could also be simply the result of a duplication without invoking selection. Whether the killing hierarchy that we observe in *P. anserina* is due to a complex battle among the *Spok* homologs or a result of the existence of the *Spok* block will require further experimentation and mathematical modeling to resolve.

## Evolutionary dynamics of the *Spok* genes

Some interesting aspects of meiotic drive in *Podospora* identified herein bear numerous features that parallel the *wtf* genes that are responsible for drive in *Schizosaccharomyces pombe*. There is no sequence similarity or conserved domains between the *Spok* and *wtf* genes, and *Podospora* and *Schizosaccharomyces* are only distantly related (~500 million years diverged) (*Wang et al., 2009*; *Prieto and Wedin, 2013*). Yet these systems display similar evolutionary dynamics within their respective species. Both of these systems are built of multiple members of gene families, which appear to duplicate, rapidly diverge to the point where they no longer show cross reactions (potentially with the aid of gene conversion), and then pseudogenize and become nonfunctional (*Nuckolls et al., 2017*; *Bravo Núñez et al., 2018a*; *Hu et al., 2017*). Both systems also have close associations with TEs (*Bowen et al., 2003*). *Hu et al. (2017)* invoke LTR-mediated non-allelic homologous recombination as a possible mechanism for *wtf* gene deletion in a lab strain of *S. pombe*. We provide evidence for the deletion of *Spok2*, but it does not fit with expectation that this deletion is LTR-mediated. Nevertheless, as TEs are still accumulating in the region, other TE-related processes may have been involved in the deletion.

The factors that determine the abundance and diversity of multigene family meiotic drivers in a species are the rates of gene duplication and loss, and time since origin. In the case of the *Spok* genes, we expect a low rate of deletion as they approach fixation because of the dikaryotic nature of *Podospora*. Specifically, when first appearing, a deletion is only expected to be present in one of the two separate nuclear genomes maintained within a dikaryon. Any selfing event should erase (i.e. drive against) the deletion, meaning that in order to become homoallelic for a deletion, the strain would have to outcross with another individual with no *Spok* genes or *Spok* genes that differ from its own. Such outcrossing could allow deletions of *Spok3* and *Spok4*, but as *Spok2* is nearly fixed in the population, any outcrossing event should also lead to the elimination of the deletion by the driving action of *Spok2*. A possible solution to the paradoxical finding that *Spok2* appears to have been lost occasionally is that the incomplete penetrance of *Spok2* may have allowed spores that were homoallelic for the deletion to survive and persist. In this sense, *Spok2* fits a model of driver turn over, wherein it is beginning to lose killing function after becoming fixed in the population. *SpokΨ1* is missing the portion of the gene that is responsible for killing and the small *Spok* fragment of *P. comata* also corresponds to the resistance part of the gene. Both of these observations suggest that the killing domain may have been lost prior to these genes becoming fully pseudogenized and hints that they may have functioned as resistance genes.

It has been pointed out that spore-killing may be a weak form of meiotic drive, because the transmission advantage is relative to the number of spores produced in a given cross, but there is no absolute increase at the population level (*Lyttle, 1991*). Hence, a spore killer was predicted by *Nauta and Hoekstra (1993)* to require an additional fitness advantage in order to reach fixation in a population. It is thus striking that *Spok2* is close to fixation in at least the European populations, bringing into question the direct fitness effects of the *Spok* genes. On the other hand, the *Spok* block (and hence *Spok3* and *Spok4*) seems to be present at relatively low frequency. It is possible that the rate at which the *Spok* block switches position is higher than the rate at which the *Spoks* can sweep to fixation. Therefore, the dynamics of *Spok* genes within the *Spok* block might differ from the *Spok2* life-cycle and might explain why spore-killing is observed to be polymorphic in *P. anserina*. In addition, *P. anserina* is capable of selfing, which may slow down the rate of fixation of the genes. Moreover, the vegetative and/or sexual expression of the *Spok* genes might be deleterious in itself, and hence natural selection might increase or maintain the frequency of strains without all *Spok* homologs. Overall, this complex system requires population genetic modeling to resolve the factors affecting the frequency of the *Spok* genes in populations of this fungus.

The relationships among the *Spok* genes can provide insight as to the evolutionary history of the *Spok* block. The observation that *Spok3* and *Spok4* are both present in the *Spok* block in a duplicate region suggests that these genes represent paralogs that formed via duplication. Indeed, the phylogenetic analysis of the UTRs agrees with a duplication origin. However, this scenario is contradicted by the finding that *Spok4* shares many features with *Spok1* of *P. comata*, but not *Spok3*. The four most likely evolutionary scenarios are outlined in *Figure 8*. If the relationship between *Spok1* and *Spok4* is a result of common descent (orthology; *Figure 8A*), then after the duplication event that generated *Spok3* and *Spok4*, *Spok3* would have to have had a much higher rate of change than

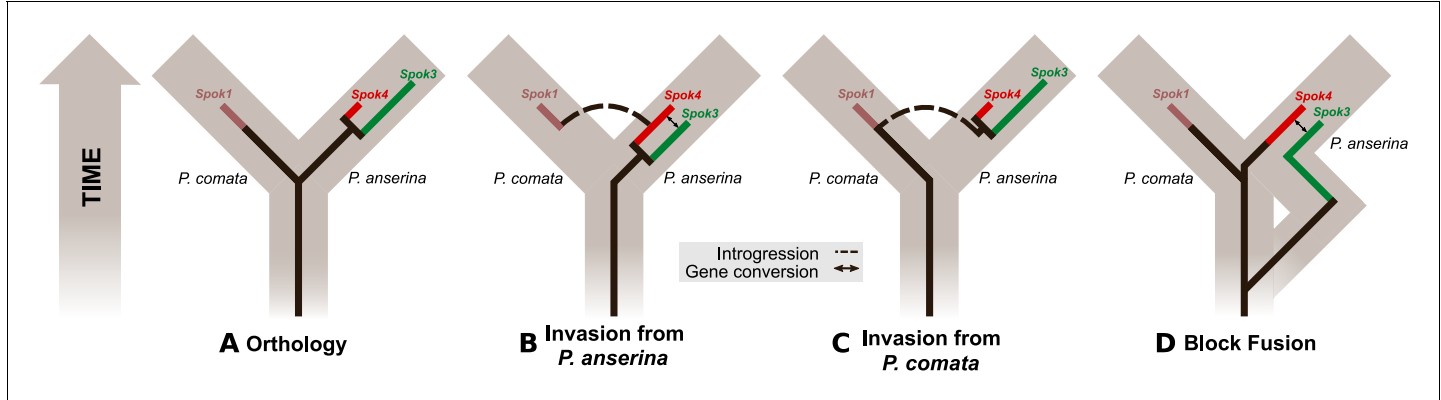

**Figure 8.** Evolutionary models of *Spok* diversification. See main text for a description of the models. The models assume that an ancestral *Spok* homolog was present in the ancestor of both *P. anserina* and *P. comata*. The *Spok* diversification is traced by the inner lines. *Spok2* is not included because its genomic location and sequence provide little clue as to its evolutionary history with respect to the other *Spok* homologs. Brown boxes represent divergence among the lineages, eventually forming *P. comata* and *P. anserina*. *P. pauciseta* is excluded for simplicity as in all cases it would need to obtain *Spok3*, *Spok4* and the *Spok* block prior to diverging from *P. anserina*, and then have no divergence since the species split, or the *Spok* block would have to be transferred between the species.

DOI: https://doi.org/10.7554/eLife.46454.033

both *Spok1* and *Spok4* in order to explain the observed divergence. In addition, the inferred deletions in the *Spok3* coding sequence would have to be reverted by gene conversion with an unknown *Spok* homolog, and new deletions would have to appear subsequently in *Spok3*.

Alternatively, the diversification of the *Spok* genes may have been influenced by past hybridization, and we discuss a few possible scenarios here. One possibility is that *Spok4* was introduced into *P. comata*, which then diverged to become *Spok1* after the duplication event (**Figure 8B**). This divergence would then have to be followed by gene conversion in order to account for the shared frameshift mutation in *Spok3* and *Spok4*. On the other hand, *Spok1* may have been transferred to *P. anserina* (**Figure 8C**), and then duplicated to form *Spok3* and *Spok4*, but this scenario requires the same additional steps as the orthology scenario. A final option is that *Spok3* and *Spok4* are not homologs formed via duplication but rather orthologs that evolved independently in separate populations (**Figure 8D**). Their current positions in the *Spok* block could be due to the fusion of ancestral blocks, followed by gene conversion. In *P. pauciseta* (CBS237.71), *Spok3*, *Spok4*, and the *Spok* block are nearly identical to the same sequences in *P. anserina*, so in order to explain the current pattern of the *Spok* genes and the *Spok* block without interspecies transfer, the block would have had to remain virtually unchanged since the divergence of *P. anserina* and *P. pauciseta*. This scenario seems highly unlikely given the relative divergence between the three species. The broader *Spok* homolog phylogeny presented here is also not consistent with a simple vertical decent model, supporting the conclusion that the evolution of the *Spok* genes has involved inter-lineage transfers. Such interspecies interactions that mediate the introgression of meiotic drive genes between species would not be a phenomenon that is unique to the *Spok* genes of *Podospora*, as meiotic drive genes in *Drosophila* have been observed to cross species boundaries and erode barriers of reproduction (**Meiklejohn et al., 2018**). Further analyses of the genomes of populations of multiple *Podospora* species is needed in order to resolve the history of the *Spok* genes and the block.

## Evolutionary history of the *Spok* gene family

**Grognet et al. (2014)** demonstrated that proteins that are related to the SPOKs are distributed across a diverse group of Ascomycota, but the majority of them are very diverged. Here, we have identified a group of more closely related homologs (clade II) from genome sequences that have been released since the **Grognet et al. (2014)** study, allowing us to analyze the evolutionary history at a finer resolution. The phylogenetic distribution of the clade II *Spok* homologs supports the general hypothesis that the *Spok* genes are transferred horizontally among evolutionarily disparate groups, as suggested by **Grognet et al. (2014)**. For example, the eurotiomycete *Polytolypa hystricis* possesses a homolog that is closely related to the *Podospora Spok* genes. However, the phylogeny

presented here shows that a subset of the clade II homologs agree with the relationships among closely related species (*Maharachchikumbura et al., 2015*), suggesting an alternative hypothesis whereby the *Spok* genes are ancestral to the Sordariomycetes but lost frequently. Such a scenario would imply that there are long-term consequences of possessing spore-killer genes, even if they are fixed in the population. These two hypotheses are not mutually exclusive, and with our data, we are not able to disentangle their relative importance for the observed pattern.

The diversification pattern may also give insight into the possibility that SPOK homologs function as drivers in lineages other than *Podospora*. The phylogeny presented here suggests that the clade I homologs do not represent meiotic drive genes because only one presumably orthologous copy is typically found. By contrast, the numerous closely related *Spok* homologs of clade II may be driving. For example, in *F. oxysporum* f. sp. *cepae* four nearly identical copies are found, resembling the distribution of the *Spok* genes in *Podospora*. However, no sexual cycle has been observed in *F. oxysporum*. Given that we demonstrate vegetative killing with *Spok3*, it is possible that the *Fusarium Spok* genes operate in vegetative tissue to ensure the maintenance of the pathogenic-associated chromosomes. Alternatively, as *F. oxysporum* strains have been found with both mating type alleles (*O'Donnell et al., 2004*), there may be a cryptic sexual cycle in which the *Spok* homologs are active.

## Conclusions

With this study, we have provided a robust connection between the phenotype and genotype of spore-killing in *P. anserina*. We showed that meiotic drive in *Podospora* spp. is governed by genes of the *Spok* family, a single locus drive system that confers both killing and resistance within a single protein, which synergize to create hierarchical dynamics by the combination of homologs at different genomic locations. We define *Psk-1*, *Psk-2*, *Psk-5*, *Psk-7*, *Psk-8*, and *Psk-S* in terms of *Spok* gene content and describe the interactions among them. The *Spok* genes are prone to duplication, diversification and movement in the genome. Furthermore, our results indicate that they probably evolved via cross-species transfer, highlighting the potential risks of the release of synthetic gene drivers for biological control invading non-target species. Moreover, we present evidence that homologs of the *Spok* genes might have similar dynamics across other groups of fungi, including pathogenic strains of *Fusarium*. Taken together, the *Spok* system provides insight into how the genome can harbor numerous independent elements that enact their own agendas and affect the evolution of multiple taxa.

## Materials and methods

### Fungal material

The fungal strains used in this study are listed in *Table 1* and were obtained from the collection maintained at the Laboratory of Genetics at Wageningen University (*van der Gaag et al., 2000*) and the University of Bordeaux. Strains with the 'Wa' identifier were collected from the area around Wageningen between 1991 and 2000 (*van der Gaag et al., 1998*; *van der Gaag et al., 2000*; *Hermanns et al., 1995*). Strains S, Y, and Z were collected in France in 1937 (*Rizet, 1952*; *Belcour et al., 1997*). Strain S is commonly used as a wildtype reference, and an annotated genome (*Espagne et al., 2008*) is publicly available at the Joint Genome Institute MycoCosm website (https://genome.jgi.doe.gov/programs/fungi/index.jsf) as 'Podan2'. A strain labeled T (referred to herein as $T_G$) was kindly provided by Andrea Hamann and Heinz Osiewacz from the Goethe University Frankfurt and originates from the laboratory of Denise Marcou. However, as the genome sequence of $T_G$ did not match that reported by *Silar et al. (2019)*, but instead is a strain of *P. anserina*, we included in our dataset another strain labeled T from the Wageningen Collection that was originally provided by the laboratory of Léon Belcour. We referred to this strain as $T_D$, and sequenced it using only Illumina HiSeq (see Appendix 2 for further discussion). It remains unclear where exactly $T_D$ and $T_G$ were collected, given the labeling confusion.

Representative strains of the *Psk* spore-killer types from the Wageningen collection were phenotyped to confirm the interactions described by *van der Gaag et al. (2000)*. Strains Wa87 and Wa53 were selected as representative of the *Psk-1* type, Wa28 for *Psk-2*, Wa21 for *Psk-3*, Wa46 for *Psk-4*, Y for *Psk-5*, Wa47 for *Psk-6*, and Wa58 for *Psk-7*. Strains S and Wa63 were used as reference strains and are annotated as *Psk-S*. Strain Wa58 mated poorly in general, so strain Z was also used as a

mating tester for the *Psk-7* spore-killer type. For all crossing experiments and genome sequencing, we isolated self-sterile monokaryons (i.e., haploid strains containing only one nuclear type) from spontaneously produced five-spored asci (*Rizet and Engelmann, 1949*), identified their mating type (*mat+* or *mat–*) by crossing them to tester strains, and annotated them with + or – signs accordingly.

## Culture and crossing conditions

All crosses were performed on Petri dishes with Henks Perfect barrage medium (HPM). This media is a modified recipe of PASM2 agar (*van Diepeningen et al., 2008*), to which 5 g/L of dried horse dung is added prior to autoclaving. Strains were first grown on solid minimal medium, PASM0.2. For each cross, a small area of mycelium of each of two monokaryons was excised from the plates and transferred to HPM. Perithecia (fruiting bodies) form at the interface between sexually compatible *mat+* and *mat–* monokaryons. Mature perithecia with fully developed ascospores were harvested after 8–11 days and the percentage of two-spored asci was evaluated to determine the killing percentage (*Box 1—figure 1*). All cultures were incubated at 27°C under 70% humidity for a 12:12 light: dark cycle. Barrage formation, whereby confrontations between mycelia of two different strains will produce a visible line of dead cells if they are vegetatively incompatible, was also evaluated on HPM. For details, see *van der Gaag et al. (2003)*.

## Experimental design for crosses

To determine the epistatic interactions between the different *Psks*, crosses were set up according to the following design (*Figure 4—figure supplement 4A*). Monokaryons of two parental strains (P1 and P2) were confronted on Petri dishes with solid HPM media and perithecia were dissected upon maturation, which takes place after 9–12 days. If only four-spored asci were observed, P1 and P2 are the same *Psk*, otherwise they represent different *Psks*. A spore was selected from a two-spored ascus to generate an F1 strain for further crosses: selfing or backcrossing to the parental strains. As most F1 strains from two-spored asci will be homokaryotic for a driver, they will result in four-spored asci when selfed, except in the case of mutual killing (*Figure 4—figure supplement 4B*). Mutual killing can also result in completely empty asci if the drivers are at the same locus. By crossing the F1 strains to both + and – strains of P1 and P2, we can distinguish between mutual killing, mutual resistance, and dominance. If none of the crosses yield two-spored asci, there is mutual resistance (*Figure 4—figure supplement 4*). In a dominance interaction, for example when P1 is dominant to P2, the F1 strain will produce four-spored asci with both mating types (+/–) of P1, but will have two-spored asci with both mating types of P2. If two-spored asci were observed in crosses to both P1 and P2, or if there are two-spored asci when the F1 is selfed, then there is mutual killing.

## DNA and RNA extraction and sequencing

### Culturing, extracting and sequencing genomic DNA using Illumina HiSeq

Monokaryotic strains of *P. anserina* were grown on plates of PASM0.2 covered with cellophane. The fungal material was harvested by scraping mycelium from the surface of the cellophane and placing 80–100 mg of mycelium in 1.5 ml Eppendorf tubes, which were then stored at −20°C. Whole-genome DNA was extracted using the Fungal/Bacterial Microprep kit (Zymo; www.zymo.com) and sequenced using the SNP and SEQ Technology platform (SciLifeLab, Uppsala, Sweden), where paired-end libraries were prepared and sequenced with the Illumina HiSeq 2500 platform (125-bp-long reads) or HiSeq X (150-bp-long reads) (*Table 1*).

### Culturing, extracting and sequencing genomic DNA using PacBio RSII

In order to generate high-molecular-weight DNA that is suitable for sequencing using PacBio, eight strains were grown on PASM0.2 for 5–7 days (*Table 1*). The agar with mycelium was cut into small pieces and used as inoculum for flasks containing 200 ml 3% malt extract solution, which were then incubated on a shaker for 10–14 days at 27°C. The mycelia were filtered from the flasks, cut into small pieces and ~1 g was allotted into 2 ml tubes with screw-on caps, after which the tubes were stored at −20°C. High-molecular-weight DNA was then extracted following the procedure described in *Sun et al. (2017)*. In brief, the mycelium was freeze-dried and then macerated, and DNA was extracted using Genomic Tip G-500 columns (Qiagen) and cleaned using the PowerClean DNA Clean-Up kit (MoBio Labs). The cleaned DNA was sequenced at the Uppsala Genome Center

(SciLifeLab, Uppsala, Sweden) using the PacBio RSII platform (Pacific Biosciences). For each sample, 10 kb libraries were prepared and sequenced using four SMRT cells and the C4 chemistry with P6 polymerase.

## Culturing, extracting and sequencing genomic DNA using MinION Oxford Nanopore

DNA extraction was performed as for the PacBio sequencing, except that the mycelia were dissected to remove the original agar inocula and the DNA was purified using magnetic beads (Speed-Beads, GE) then sequenced without further size-selection. Monokaryotic samples $T_G+$ and CBS237.71– were sequenced first in a barcoded run on a R9.5.1 flowcell using the Oxford Nanopore Technologies (ONT) rapid barcoding kit (1.5 µl RBK004 enzyme to 8.5 µl DNA per reaction). Owing to low tagmentation efficiency, we did additional sequencing for $T_G+$ using the ligation sequencing kit (LSK108, R9.4.1 flowcell). 500 ng DNA (25 µl) was mixed with 1.5 µl NEB Ultra-II EP enzyme and 3.5 µl NEB Ultra-II EP buffer and incubated for 10 min at 20°C and 10 min at 65°C, before addition of 20 µl AMX adaptor, 1 µl ligation enhancer, and 40 µl NEB Ultra-II ligase. After ligation, the standard ONT washing and library loading protocol was followed and the sample was sequenced on a R9.4.1 flowcell. After sufficient sequencing depth had been achieved for sample $T_G+$, the flowcell was washed and the remaining barcoded samples were loaded to improve coverage for sample CBS237.71–. The sample Y+ yielded less DNA (150 ng in 15 µl) and hence half the normal volume of adaptor was used (10 µl) and ligated using 20 µl Blunt/TA ligase for 15 min. Otherwise, the standard protocol was followed, with sequencing done in a R9.4.1 flowcell. Basecalling and barcode split was done using Guppy 1.6 and Porechop (ONT) for all samples.

## RNA sequencing

We generated transcriptomic data from dikaryotic strains that undergo spore-killing during selfing. The $S_{14}$ backcrosses (see below) were mated to the strain S in order to obtain killer heteroallelic spores (from four-spore asci) that were dissected from ripe fruiting bodies (see *Figure 2—figure supplement 2*). The spores were germinated in plates of PASM2 with 5 g/L ammonium acetate added. Two days after germination, the culture was stored in PASM0.2 media at 4°C to arrest growth. From that stock, we inoculated HPM plates with either a polycarbonate track etched 76 mm 0.1 µm membrane disk (Poretics, GVS Life Sciences, USA) (Psk1x$S_5$ and Psk7x$S_5$) or a cellophane layer (Psk2x$S_5$ and Psk5x$S_5$) on top. The mycelia were grown for ~11 days and harvested for RNA extraction when the first spores were shot into the plate lid, ensuring several stages of fruiting body development. Note that *P. anserina* starts to degrade cellophane after ~6 days, and therefore the polycarbonate membrane allows for longer growing periods. Spore-killing was independently confirmed on HPM plates inoculated without a membrane. In addition, in order to improve gene annotation, we grew the strains Wa63– and $T_D+$ on a cellophane layer on HPM for 11 and 7 days, respectively, to capture transcripts occurring during the monokaryotic phase.

The harvested mycelia were immediately frozen in liquid nitrogen and stored at −80°C until RNA extraction. Next, 150 mg of frozen tissue was ground under liquid nitrogen and total RNA was extracted using an RNeasy Plant Mini Kit (Qiagen, Hilden, Germany). The quality of RNA was checked on the Agilent 2100 Bioanalyzer (Agilent Technologies, USA). All RNA samples were treated with DNaseI (Thermo Scientific). Sequencing libraries were prepared using an NEBNext Ultra Directional RNA Library Prep Kit for Illumina (New England Biolabs). The mRNA was selected by purifying polyA+ transcripts (NEBNext Poly(A) mRNA Magnetic Isolation Module, New England Biolabs). Finally, paired-end libraries were sequenced with Illumina HiSeq 2500 at the SNP and SEQ Technology platform.

## Genomic analyses

For both DNA and RNA Illumina HiSeq reads, adapters were identified with cutadapt v. 1.13 (*Martin, 2011*) and then trimmed using Trimmomatic 0.36 (*Bolger et al., 2014*) with the options ILLUMINACLIP:adapters.fasta:1:30:9 LEADING:20 TRAILING:20 SLIDINGWINDOW:4:20 MINLEN:30. Only filtered reads with both forward and reverse were kept for downstream analyses. For short-read mapping, we used BWA v. 0.7.17 (*Li and Durbin, 2010*) with PCR duplicate marking of Picard v. 2.18.11 (http://broadinstitute.github.io/picard/), followed by local indel re-aligning implemented in

the Genome Analysis Toolkit (GATK) v. 3.7 (*Van der Auwera et al., 2013*). Mean depth of coverage was calculated with QualiMap v.2.2 (*Okonechnikov et al., 2016*).

The raw PacBio reads were filtered and assembled with the SMRT Analysis package and the HGAP 3.0 assembler (*Chin et al., 2013*). The resulting assembly was error-corrected (polished) with Pilon v. 1.17 (*Walker et al., 2014*) using the mapped filtered Illumina reads of the same monokaryotic strain. The samples sequenced with MinION were assembled using Minimap2 v. 2.11 (*Li, 2018*) and Miniasm v. 0.2 (*Li, 2018*; *Li, 2016*), polished twice with Racon v. 1.3.1 (*Vaser et al., 2017*) using the MinION reads, and further polished for five consecutive rounds of Pilon v. 1.22 using the Illumina reads as above. Scaffolds were assigned to chromosome numbers on the basis of homology with Podan2. Small scaffolds (<100 kb) corresponding to rDNA and mitochondrial-derivatives were discarded. Only the biggest mitochondrial scaffold was retained. In addition, DNA Illumina reads were assembled de novo for each sample using SPAdes v. 3.12.0 (*Bankevich et al., 2012*; *Antipov et al., 2016*) using the k-mers 21,33,55,77 and the –careful option. BLAST searches of the scaffolds in the final assembly of the strain CBS237.71 revealed contamination by a *Methylobacterium* sp. in the MinION data (but not in the Illumina data set). The scaffolds matching the bacterium were removed from the analysis. Long-read assemblies were evaluated using BUSCO v. 3.0.2 (*Simão et al., 2015*; *Waterhouse et al., 2017*) for the Sordariomyceta ortholog set with the following dependencies: BLAST suit 2.6.0+ (*Camacho et al., 2009*), HMMER v, 3.1b2 (*Mistry et al., 2013*), and AUGUSTUS v 3.2.3 (*Stanke and Waack, 2003*). Short-read assemblies were evaluated using QUAST v. 4.6.3 (*Mikheenko et al., 2016*).

The assembly of the *Spok* block was visually inspected by mapping the long reads (using Minimap2) and the short reads (BWA) as above into the long-read polished assemblies. As the MinION assemblies maintain some degree of sequencing error at repetitive regions that cannot be confidentially polished, we also assembled both types of reads into a hybrid assembly using SPAdes (same options as above) and, whenever different for short indels or SNPs but fully assembled, the sequence of the *Spok* genes was taken from the (low-error) hybrid assembly. Assembly of the *Spok* block of the $T_G$+ strain was particularly challenging because the recovered MinION reads were relatively short. However, a few (<10) reads were long enough to cover the tandem duplication that contains *Spok3* (albeit with high nucleotide error rate in the assembly). The hybrid SPAdes assembly collapsed the duplication into a single copy. We therefore mapped the short reads into the hybrid assembly, confirming that the *Spok3* gene had doubled coverage and no SNPs, as expected for a perfect duplication.

Alignments of the assembled genomes were performed with the NUCmer program of the MUMmer package v. 4.0.0beta2 (*Kurtz et al., 2004*) using options –b 200 c 2000 –maxmatch, except when otherwise noted. The figures showing alignments of chromosome 5, the *Spok* block, and the *Spok2* region (*Figure 3*, *Figure 5*, and *Figure 5—figure supplements 1* and *2*) were generated by extracting the regions from each de novo assembly and aligning them in a pairwise fashion. The NUCmer output was then visualized using a custom Python script. The distribution of GC was plotted along the chromosomes with a custom Python script using 4-kb windows and steps of 2 kb.

In order to evaluate synteny across species, we aligned the chromosomal scaffolds of Wa58– (*P. anserina*, Psk-7), CBS237.71- (*P. pauciseta*), and the reference genome of *P. comata* (PODCO, *Silar et al., 2019*) using NUCmer with –b 2000 in an all-vs-all fashion. Given that the largest TE reported for *P. anserina* is around 12 kb (*Espagne et al., 2008*), we filtered out alignments smaller than 13 kb and plotted the remaining alignments using Circos (*Krzywinski et al., 2009*). We further excluded alignments of missing data (Ns) tracks in chromosome 6 and 7 from PODCO. See https://github.com/johannessonlab/SpokPaper (*Ament-Velásquez, 2019*; copy archived at https://github.com/elifesciences-publications/SpokPaper) for a Snakemake pipeline and Circos configuration files. Notice that the Circos plot includes intrachromosomal alignments; for example, the rDNA operon in chromosome 3 is especially noticeable for Wa58–. To evaluate the large-scale translocations in *P. comata* (*Figure 1B* and *Figure 1—figure supplement 1*), we mapped the short-reads of $T_D$+ to PODCO and Podan2, inferring mis-assemblies on the basis of the concordance of paired-end reads. Translocation 1 is clearly a misassembly. The translocation 2 in chromosome 4 however is complex because the corresponding boundaries in Podan2 start at a cluster of TEs at the 5′ end, and finish at the centromere on the 3′ end. Indeed, *Silar et al. (2019)* could not verify this translocation using PCR. Accordingly, the mapping of the paired-end reads does not support the translocation to the end of chromosome 4 in PODCO.

## Genome annotation

For annotation, we opted for gene prediction trained specifically on *P. anserina* genome features. We used the ab initio gene prediction programs GeneMark-ES v. 4.32 (*Lomsadze et al., 2005*; *Ter-Hovhannisyan et al., 2008*) and SNAP release 2013-06-16 (*Korf, 2004*). All of the training process was performed on the sample Wa28–, for which all chromosomes were assembled (see 'Results'). The program GeneMark-ES was self-trained with the script `gmes_petap.pl` and the options `–fungal –max_intron 3000 min_gene_prediction 120`. SNAP was trained as instructed in the tutorial of the MAKER pipeline v. 2.31.8 (*Holt and Yandell, 2011*; *Campbell et al., 2014*, and in the SNAP README file). First, we use the Podan2 transcripts and protein models as sole evidence to infer genes with MAKER (option est2genome = 1) and then we had a first round of SNAP training. The resulting HMM file was used to re-run MAKER (est2genome = 0) and to re-train SNAP, obtaining the final HMM training files.

A library of repetitive elements was constructed by collecting the reference *P. anserina* TEs described in *Espagne et al. (2008)* available in Genbank, and combining them with the fungal portion of Repbase version 20170127 (*Bao et al., 2015*) and the *Neurospora* library of *Gioti et al. (2013)*. In order to produce transcript models, we used STAR v. 2.6.1b (*Dobin et al., 2013*) with maximum intron length of 1000 bp to map the RNAseq reads of all samples, followed by processing with Cufflinks v. 2.2.1 (*Trapnell et al., 2010*). For the final genome annotation, we used MAKER v. 3.01.02 along with GeneMark-ES v. 4.33, SNAP release 2013-11-29, RepeatMasker v. 4.0.7 (http://www.repeatmasker.org/), BLAST suit 2.6.0+ (*Camacho et al., 2009*), Exonerate v. 2.2.0 (*Slater and Birney, 2005*), and tRNAscan-SE v. 1.3.1 (*Lowe and Eddy, 1997*). After preliminary testing, we chose the transcripts of $Psk7xS_{14}$ (mapped to the PacBio assembly of Wa58– ) and Wa63– (PacBio assembly of the same strain) as expressed sequence tag (EST) evidence, and the Podan2 and $T_D$ (*Silar et al., 2019*) models as protein evidence. The MAKER models of relevant regions were manually curated by comparing with RNAseq mapping and coding sequences (CDS) produced with Trans-Decoder v. 5.5.0 (*Haas et al., 2013*) on the Cufflinks models.

We used the blastn program to localize possible copies of *Spok* genes in all genome assemblies. The *Spok2* (Pa_5_10) gene from *Grognet et al. (2014)* was selected as the query. We named the new *Spok* genes (*Spok3* and *Spok4*) arbitrarily on the basis of sequence similarity, as reflected in the phylogenetic analyses (see below). Note that the existence of *Spok3* had previously been hypothesized by *Grognet et al. (2014)*, but no DNA sequence was provided. Moreover, the strain Y, in which they identified *Spok3*, contains both *Spok3* and *Spok4*.

## Introgressions of the spore-killing loci

Backcrossed strains of the various spore-killer phenotypes were generated through five recurrent backcrosses to the reference strain S ($S_5$) by *van der Gaag et al. (2000)* . In the original study, the strains selected as spore-killer parents were Wa53+ for *Psk-1*, Wa28– for *Psk-2*, Y+ for *Psk-5*, and Wa58– for *Psk-7*. The $S_5$ strains are annotated as Wa170 (*Psk-1*), Wa130 (*Psk-2*), Wa200 (*Psk-5*), and Wa180 (*Psk-7*) in the Wageningen collection, but for the sake of clarity, we refer to them as $Psk1xS_5$, $Psk2xS_5$, $Psk5xS_5$, and $Psk7xS_5$.

We sequenced the $S_5$ strains along with the reported parental strains using Illumina HiSeq 2500. We mapped the reads to Podan2 as described above, and then performed SNP calling using the HaplotypeCaller pipeline of GATK (options: –ploidy 1 –newQual –stand_call_conf 20.0). We removed sites that had missing data, that overlapped with repeated elements as defined by RepeatMasker, or in which all samples were different from the reference genome, using VCFtools v. 0.1.16 (*Danecek et al., 2011*), BEDtools v. 2.27.1 (*Quinlan and Hall, 2010*), and BCFtools v. 1.9 (*Danecek and McCarthy, 2017*), respectively. We plotted the density of filtered SNPs across the genome with the R packages vcfR (*Knaus and Grünwald, 2017*; *Kamvar et al., 2015*) and poppr (*Knaus and Grünwald, 2017*; *Kamvar et al., 2015*). A full Snakemake (*Johannes and Rahmann, 2018*) pipeline can be found at https://github.com/johannessonlab/SpokPaper. Notice that we sequenced both monokaryons of our strain S to account for the mutations that might have had occurred since the separation of the reference S strain in the laboratory of *Espagne et al. (2008)* and our S strain from the Wageningen collection. These mutations should be present in the backcrosses, but they are independent from the spore=killer elements.

Inspection of the introgressed tracks revealed that the variants of the backcross Psk1xS$_5$ do not match Wa53+ (the reported parent) perfectly. Given that the *Spok* content is the same as that in Wa53+, the introgressed track co-occurs with the expected position of the *Spok* block on chromosome 3, and the fact that the phenotype of this backcross matches a *Psk-1* spore-killer type, we concluded that Wa170 (Psk1xS$_5$) in the collection actually belongs to another of the *Psk-1* backcrosses described in the doctoral thesis of *van der Gaag (2005)*, probably backcrossed from Wa52. Puzzlingly, an introgressed track in the chromosome 3 of the Psk2xS$_5$ strain does not match the expected parent (Wa28) either, both in SNPs and *het*-gene alleles *Figure 4—figure supplement 3*). However, other tracks in different chromosomes, including that of chromosome 5 where the *Psk-2 Spok* block can be found, do match Wa28. Likewise, like Wa28, Psk2xS$_5$ only has *Spok2* and *Spok3* copies. Hence, we concluded that our results are not affected by these inconsistencies.

As reported by *van der Gaag et al. (2000)*, the S$_5$ strains were generated by selecting ascospores from two-spored asci of crosses between S and the spore killer parent. This procedure ensures that the offspring will be homozygous for alleles of the spore-killer parent from the spore-killing locus to the centromere (*Box 1—figure 1* and *Figure 4—figure supplement 3*). To eliminate as much background as possible from the spore killer parents in the backcrossed strains, nine additional backcrosses were conducted where ascospores were selected from four-spored and two-spored asci in alternating generations. Ascospores from the final generation were selected from two-spored asci to ensure that the strains would be homozygous at the spore-killing locus. These strains are the result of 14 backcrosses to S (S$_{14}$) and are annotated as Psk1xS$_{14}$, Psk2xS$_{14}$, Psk5xS$_{14}$, and Psk7xS$_{14}$. The S$_5$ and S$_{14}$ strains were phenotyped by crossing the strains to their parents as well as other reference spore-killer strains to confirm that the killing phenotypes remained unchanged after the backcrosses.

## Knock-out of *Spok2*

To knock-out *Spok2*, a 459-bp and a 495-bp fragment flanking the *Spok2* ORF downstream and upstream were obtained by PCR. These fragments were then cloned flanking the *hph* gene in the SKhph plasmid as blunt end fragments in a *Eco*RV site and a *Sma*I site. The deletion cassette was then amplified by PCR and used to transform a ΔKu70 strain (*El-Khoury et al., 2008*). Five transformants were screened for integration of the *hph* marker at *Spok2* by PCR and crossed to s. To purify the Δ*Spok2* nuclei, a heterokaryotic binucleated Δ*Spok2* spore was recovered from a two-spored ascus and used to fertilize the initial Δ*Spok2* transformant (which may or may not be heterokaryotic). Uninucleated *hygR*-resistant spores were then recovered from this cross.

## Construction of a disruption cassette to insert *Spok3* or *Spok4* into the *PaPKS1* locus

To replace the ORF of the centromere-linked Pa_2_510 (*PaPKS1*) gene by one of the *Spok3* or *Spok4* genes (see 'Results'), a disruption cassette was constructed as follows. A DNA fragment corresponding to the 700-bp upstream region of the *PaPKS1* ORF was amplified with oligonucleotides UpPks1_F and UpPks1_R. This fragment was then cloned into a SKpBluescript vector (Stratagene) containing the nourseothricin-resistance gene *Nat* in the *Eco*RV site (vector named P1) using the *Sac*II/*Not*I restriction enzymes (upstream from the *Nat* gene) to produce the P1UpstreamPKS1 vector. Then, the 770 bp downstream of the *PaPKS1* ORF was amplified with oligonucleotides DownPks1_F and DownPks1_R. This second fragment was cloned into the P1UpstreamPKS1 vector using the *Hind*III/*Sal*I restriction enzymes (downstream from the *Nat* gene) to produce the P1UpstreamDownstreamPKS1 vector. Finally, *Spok3* was amplified from the Wa28 strain with oligonucleotides UpSpok3 and 4_F and DownSpok3, and *Spok4* was amplified from the Wa87 strain with oligonucleotides UpSpok3 and 4_F and DownSpok4. These genes were then cloned into the P1UpstreamDownstreamPKS1 vector using the *Not*I/*Xba*I restriction enzymes (between *PaPKS1* upstream region and *Nat* gene) to produce the P1UpstreamDownstreamPKS1_Spok3 or the P1UpstreamDownstreamPKS1_Spok4 vector, so that the *Spok3* or *Spok4* and *Nat* genes are flanked by the upstream and downstream regions of *PaPKS1* ORF, allowing *PaPKS1* ORF replacement by homologous recombination. The *Spok3* and *Spok4* amplified *Spok* genes contain the ORFs flanked by the 983-bp upstream the start codon and the 460-bp downstream the stop codon for *Spok3*, and by the 984-bp upstream the start codon and the 393-bp downstream the stop codon for *Spok4*,

allowing the expression of the *Spok* genes using their native promoter and terminator regions. The disruption cassettes were then amplified from the final vectors using the most distal oligonucleotides 3′insidePsk1_F and 3′insidePsk1_R and named PKS1::Spok3_nat-1 and PKS1::Spok4_nat-1. See ***Supplementary file 4*** for primer sequences. Point mutations in the *Spok3* gene were obtained by site-directed mutagenesis using the Q5 high-fidelity polymerase (New England Biolabs) and verified by Sanger sequencing.

The *P. anserina* Δ*Spok2* (ΔPa_5_10) strain was obtained after disruption of the gene Pa_5_10 and replacement of its ORF with the hygromycin-resistance gene *hph* in a ΔKu70 strain. This strain was used as recipient strain for the disruption cassettes. We used 5 µl of the cassettes for transfection and Nourseothricin-resistant transformants were selected. As expected, most of the transformants were unpigmented and corresponded to the insertion of *Spok3* or *Spok4* by replacement of *PaPKS1*. Gene replacement was verified by PCR.

## Protein-annotation methods

Prediction of unstructured regions was performed in SPOK3 with PrDOS with a 2% false-positive setting (***Ishida and Kinoshita, 2007***). Coiled-coil prediction was performed with LOGICOIL (***Vincent et al., 2013***), CCHMM_PROF (***Bartoli et al., 2009***) and Multicoil2 (***Wolf et al., 1997***). Domain prediction was performed using Gremlin (***Balakrishnan et al., 2011***) and RaptorX contact predict (***Ma et al., 2015***). Conserved residues were identified using Weblogo 3 (***Crooks et al., 2004***) with a Gremlin-generated alignment as input. Domain identification was done with HHPred (***Zimmermann et al., 2018***).

In order to compare the diversity at the nucleotide level with the protein models, we calculated the average pairwise nucleotide differences (***Nei and Li, 1979***) for each bi-allelic site (correcting by the number of sites (n/(n −1)) while ignoring sites with gaps) on a *Spok* alignment (see below), using overlapping windows of 100 bp and steps of 20 bp. This procedure was performed on a selected representative of each *Spok* homolog (*Spok2* of S, *Spok3* and *Spok4* of Wa87, and *Spok1* from $T_D$), or for all the alleles of each *Spok* within the *P. anserina* strains. Values of $d_N/d_s$ were calculated using the seqinr package in R (***Charif and Lobry, 2007***).

In order to infer the relationship between the three *Podospora* taxa investigated herein, we used OrthoFinder v. 2.2.6 (***Emms and Kelly, 2015***; ***Emms and Kelly, 2019***) to define orthogroups across all samples with long-read data and the reference genomes of *P. anserina* (***Espagne et al., 2008***) and *P. comata* (***Silar et al., 2019***). We randomly selected 1000 orthogroups of single-copy orthologs. As OrthoFinder works with protein sequence, we used the Podan2 ortholog to do BLAST searches against each genome and extracted the corresponding best hit as nucleotide sequence (including introns). We then aligned each orthogroup using MAFFT v. 7.407 (***Katoh et al., 2017***) with options `—maxiterate 1000 —retree 1 —localpair`. We concatenated the 1000 alignments (1,652,987 sites, from which 27,000 were variable) and used the resulting matrix as input for SplitsTree4 v. 4.14.16, build 26 Sep 2017 (***Huson and Bryant, 2006***) to construct an unrooted split network with a NeighborNet (***Bryant and Moulton, 2002***) distance transformation (uncorrected distances) and an EqualAngle splits transformation. To estimate the genetic (gene) distance between species, we averaged the identity across all *P. anserina* samples vs *P. pauciseta* (CBS237.71) and vs. *P. comata* (***Silar et al., 2019***), using the values produced in the distance matrix of SplitsTree.

The final gene models of all the *Spok* genes in *Podospora* spp. were aligned along with the sequences of *Spok2* and *Spok1* from ***Grognet et al. (2014)*** using MAFFT online version 7 (***Katoh et al., 2017***) with default settings (only one copy of *Spok3* from $T_G$ was used). The resulting alignment was manually corrected taking into account the reading frame of the protein. As the UTRs seem to be conserved between paralogs, 654 (5′ end) and 250 (3′ end) bp of the flanking regions with respect to *Spok2* were also included in the alignment. An unrooted split network was constructed in SplitsTree4 as above. SplitsTree4 was used likewise to perform a Phi test for recombination (***Bruen et al., 2006***), using a windows size of 100 and k = 6. In addition, we used the BlackBox of RAxML-NG v. 0.6.0 (***Kozlov et al., 2019***) to infer maximum likelihood phylogenetic trees of the nucleotide alignment of the 5′ UTR, the coding sequence (CDS), and the 3′ UTR of the *Spok* homologs. We ran RAxML-NG with 10 parsimony and 10 random starting trees, a GTR +GAMMA (four categories) substitution model, and 100 bootstrap pseudo-replicates for each analysis.

In order to create a phylogeny of proteins that are closely related to the products of the *Spok* genes in *Podospora* (and hence likely to be meiotic drivers), the protein sequence of *Spok1* was

used as a query against the NCBI genome database (as of 14 November 2018). We collated all hits with e-values lower than Fs_82228, which has been shown previously to have some spore-killing functionality in *P. anserina* (*Grognet et al., 2014*), with hit coverage greater than 75% and no missing data (Ns) in the sequence. The sequences were aligned using the codon-aware program MACSE v. 2.03 (*Ranwez et al., 2018*), with the representative *Podospora Spok* genes set as 'reliable' sequences (–seq), and the rest as 'non reliable' (–seq_lr). Many of the original gene models predict introns in the sequences, but no divergent regions were apparent in the alignment and, even if present, MACSE tends to introduce compensatory frame shifts. The entire gene alignment was used for the analysis. The resulting nucleotide alignment was corrected manually, translated into amino acids, and trimmed with TrimAl v. 1.4.1 (*Capella-Gutiérrez et al., 2009*) using the gappyout function. A maximum likelihood tree was then produced using IQ-TREE v. 1.6.8 (*Kalyaanamoorthy et al., 2017*; *Nguyen et al., 2014*) with extended model selection (–m MFP) and 1000 standard bootstrap pseudo-replicates. The protein sequence Uv_5543 of *Ustilaginoidea virens* was selected as an outgroup on the basis of a BioNJ tree made with SeaView v. 4.5.4 (*Gouy et al., 2010*) of the Gremlin alignment described above.

For comparison, we performed a phylogenomic analysis of all of the strains that had at least one BLAST hit for the *Podospora Spok* homologs, as defined above. Briefly, we recovered all the protein sequences for each genome from GenBank and ran OrthoFinder to recover orthogroups. We obtained 288 single-copy orthogroups that were then processed with PREQUAL v.1.02 (*Whelan et al., 2018*) and aligned with MAFFT. Columns with more than 50% missing data were removed with TrimAl (–gt 0.5) and all alignments were concatenated. The supermatrix was analyzed with IQTree as above but with 100 standard bootstraps.

## Pool-sequencing of *Psk-1* vs *Psk-5* progeny

In order to confirm that *Spok2* is responsible of the killing relationship between *Psk-5* and *Psk-1*, we conducted a cross between the strains Wa87 and Y. When perithecia started shooting spores, we replaced the lid of the cross plate with a water-agar plate upside-down, and let it sit for around an hour. As the *P. anserina* spores from a single ascus typically land together, it is possible to distinguish spores that came from an ascus with no killing (groups of four spores) from those that survived killing (groups of two spores). To improve germination rates, we scooped spore groups of the same ascus type and deposited them together in a single plate of germination medium. After colonies became visible, they were transferred into a PASM2 plate with a cellophane layer where they grew until DNA extraction, which was followed by pool-sequencing with Illumina HiSeq X. In total, 21 two-spore groups, and 63 four-spore groups were recovered.

The resulting short reads were quality controlled and mapped to Podan2 as above. We used GATK to call variants from the parental strains (treated as haploid) and the two pool-sequencing databases (as diploids). We then extracted SNPs, removed sites with missing data, and attempted to quantify the coverage frequency of the parental genotypes for each variant. The expectation was that spore killing (two-spore asci) would result in a long track of homozygosity (only one parental genotype) around *Spok2*, as compared to the fully heterozygous four-spore asci. A full Snakemake pipeline is available at https://github.com/johannessonlab/SpokPaper.

## Acknowledgements

We would like to thank Magdalena Grudzinska-Sterno for valuable assistance with DNA and RNA extractions as well as library preparations. We are also thankful to Ola Wallerman for assistance with MinION Oxford Nanopore sequencing. We thank Mathieu Paoletti for discussion and access to RNA-seq pilot data at early stages of the project. We acknowledge the support of the National Genomics Infrastructure (NGI)/Uppsala Genome Center for assistance with massive parallel sequencing. The computations were performed on resources provided by SNIC through the Uppsala Multidisciplinary Center for Advanced Computational Science (UPPMAX) under Project SNIC 2017/1-567. This study was founded by a European Research Council grant under the program H2020, ERC-2014-CoG, project 648143 (SpoKiGen), with additional funding from The Swedish Research Council (VR) (to HJ), and from the Lars Hierta Memorial Foundation and The Nilsson-Ehle Endowments of the Royal Physiographic Society of Lund (to SLAV).

# Additional information

## Funding

| Funder | Grant reference number | Author |
|---|---|---|
| H2020 European Research Council | ERC-2014-CoG | Hanna Johannesson |
| Swedish Research Council | Spore killer genomics | Hanna Johannesson |
| Lars Hierta Memorial Foundation | Podospora genomics | S Lorena Ament-Velásquez |
| Royal Physiographic Society in Lund | Podospora genomics | S Lorena Ament-Velásquez |

The funders had no role in study design, data collection and interpretation, or the decision to submit the work for publication.

## Author contributions

Aaron A Vogan, Conceptualization, Formal analysis, Validation, Investigation, Visualization, Methodology, Writing—original draft; S Lorena Ament-Velásquez, Conceptualization, Data curation, Software, Formal analysis, Funding acquisition, Validation, Investigation, Visualization, Methodology, Writing—review and editing; Alexandra Granger-Farbos, Virginie Coustou, Hélène Yvanne, Investigation; Jesper Svedberg, Conceptualization, Investigation, Visualization, Writing—review and editing; Eric Bastiaans, Conceptualization, Investigation, Writing—review and editing; Alfons JM Debets, Conceptualization, Resources, Writing—review and editing; Corinne Clavé, Supervision, Investigation, Writing—review and editing; Sven J Saupe, Conceptualization, Formal analysis, Supervision, Investigation, Visualization, Methodology, Writing—review and editing; Hanna Johannesson, Conceptualization, Supervision, Funding acquisition, Project administration, Writing—review and editing

## Author ORCIDs

Aaron A Vogan ![iD] https://orcid.org/0000-0003-2013-7445
S Lorena Ament-Velásquez ![iD] https://orcid.org/0000-0003-3371-9292
Hanna Johannesson ![iD] https://orcid.org/0000-0001-6359-9856

## Decision letter and Author response

Decision letter https://doi.org/10.7554/eLife.46454.059
Author response https://doi.org/10.7554/eLife.46454.060

# Additional files

## Supplementary files

• Supplementary file 1. Statistics from PacBio and Nanopore assemblies.
DOI: https://doi.org/10.7554/eLife.46454.034

• Supplementary file 2. Statistics from SPAdes assemblies. As a proxy of completeness, the coverage of the Podan2 reference genome is given, as is the mean depth of coverage to Podan2 as reported by Qualimap. n, number of scaffolds; n500, number of scaffolds larger than 500 bp; min, size of smaller scaffold; max, size of largest scaffold; sum_n:500, sum of the length of all scaffolds larger than 500 bp; sum, size of the assembly.
DOI: https://doi.org/10.7554/eLife.46454.035

• Supplementary file 3. *Spok* gene content of genetically modified strains.
DOI: https://doi.org/10.7554/eLife.46454.036

• Supplementary file 4. Primers used in this study.
DOI: https://doi.org/10.7554/eLife.46454.037

• Transparent reporting form
DOI: https://doi.org/10.7554/eLife.46454.038

## Data availability

The full CDS sequence and UTRs of Spok3, Spok4, and SpokΨ1 (strain Wa87+) were deposited in NCBI GenBank under the accession numbers MK521588, MK521589, and MK521590, respectively. Raw sequencing reads were deposited on the NCBI SRA archive under the BioProject PRJNA523441. Final assemblies of samples with long-read data and alignment files (in fasta and nexus format) used to produce the main figures are available in the Dryad Digital Repository (https://doi.org/10.5061/dryad.vm1192g). Scripts and snakemake pipelines can be found at https://github.com/johannessonlab/SpokPaper (copy archived at https://github.com/elifesciences-publications/SpokPaper).

The following datasets were generated:

| Author(s) | Year | Dataset title | Dataset URL | Database and Identifier |
|---|---|---|---|---|
| Vogan AA, Ament-Velásquez SL, Granger-Farbos A, Svedberg J, Bastiaans E, Debets AJM, Coustou V, Yvanne H, Clavé C, Saupe SJ, Johannesson H | 2019 | Data from: Combinations of Spok genes create multiple meiotic drivers in Podospora | https://doi.org/10.5061/dryad.vm1192g | Dryad Digital Repository, 10.5061/dryad.vm1192g |
| Vogan AA, Ament-Velásquez SL, Granger-Farbos A, Svedberg J, Bastiaans E, Debets AJM, Coustou V, Yvanne H, Clavé C, Saupe SJ, Johannesson H | 2019 | Full CDS sequence and UTRs of Spok3, Spok4, and Spok1 (strain Wa87+) | https://www.ncbi.nlm.nih.gov/nuccore/MK521588 | NCBI GenBank, MK521588 |
| Vogan AA, Ament-Velásquez SL, Granger-Farbos A, Svedberg J, Bastiaans E, Debets AJM, Coustou V, Yvanne H, Clavé C, Saupe SJ, Johannesson H | 2019 | Full CDS sequence and UTRs of Spok3, Spok4, and Spok1 (strain Wa87+) | https://www.ncbi.nlm.nih.gov/nuccore/MK521589 | NCBI GenBank, MK521589 |
| Vogan AA, Ament-Velásquez SL, Granger-Farbos A, Svedberg J, Bastiaans E, Debets AJM, Coustou V, Yvanne H, Clavé C, Saupe SJ, Johannesson H | 2019 | Full CDS sequence and UTRs of Spok3, Spok4, and Spok1 (strain Wa87+) | https://www.ncbi.nlm.nih.gov/nuccore/MK521590 | NCBI GenBank, MK521590 |
| Vogan AA, Ament-Velásquez SL, Granger-Farbos A, Svedberg J, Bastiaans E, Debets AJM, Coustou V, Yvanne H, Clavé C, Saupe SJ, Johannesson H | 2019 | Raw sequencing reads | https://www.ncbi.nlm.nih.gov/bioproject/PRJNA523441 | NCBI Sequence Read Archive, PRJNA523441 |

The following previously published datasets were used:

| Author(s) | Year | Dataset title | Dataset URL | Database and Identifier |
|---|---|---|---|---|
| Espagne E, Lespinet O, Malagnac F, Da Silva C, Jaillon | 2008 | Podan2 | https://www.ebi.ac.uk/ena/data/view/GCA_000226545.1 | European Nucleotide Archive, GCA_000226545.1 |

| | | | | | |
|---|---|---|---|---|---|
| O, Porcel BM, Couloux A, Aury JM, Ségurens B, Poulain J, Anthouard V, Grossetete S, Khalili H, Coppin E, Déquard-Chablat M, Picard M, Contamine V, Arnaise S, Bourdais A, Berteaux-Lecellier V, Gautheret D, de Vries RP, Battaglia E, Coutinho PM, Danchin EG, Henrissat B, Khoury RE, Sainsard-Chanet A, Boivin A, Pinan-Lucarré B, Sellem CH, Debuchy R, Wincker P, Weissenbach J, Silar P | | | | | |
| Silar P, Dauget JM, Gautier V, Grognet P, Chablat M, Hermann-Le Denmat S, Couloux A, Wincker P, Debuchy R | 2019 | PODCO | https://www.ebi.ac.uk/ena/data/view/GCA_900290415.1 | | European Nucleotide Archive, GCA_900290415.1 |

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

## Appendix 1

DOI: https://doi.org/10.7554/eLife.46454.004

### The biology of *Podospora*

The life cycle of *P. anserina* is an important factor to consider when discussing the meiotic drive of the *Spok* genes. Although it has haploid nuclei, *P. anserina* maintains a dikaryotic (n + n) state throughout its entire lifecycle. Haploid nuclei of different mating-types are shown as white and black circles within fungal cells. The fruiting body (perithecium) is generated from dikaryotic (n + n) mycelia, usually from a single individual strain. Within the perithecium, the sexual cycle is completed to produce four dikaryotic ascospores per ascus. Occasionally, atypical spore formation may occur and can result in the production of five spores in an ascus, of which two are small and monokaryotic (n). These are self-sterile and need to outcross either with a monokaryotic individual of the opposite mating type or with a dikaryotic individual to complete the life cycle. Note that outcrossing may occur via mating between either siblings or unrelated individuals of the opposite mating type. The monokaryotic spores are useful for generating self-sterile (haploid) cultures for the purposes of sequencing and laboratory mating and are used to generate the + and -strains that are discussed in the main text.

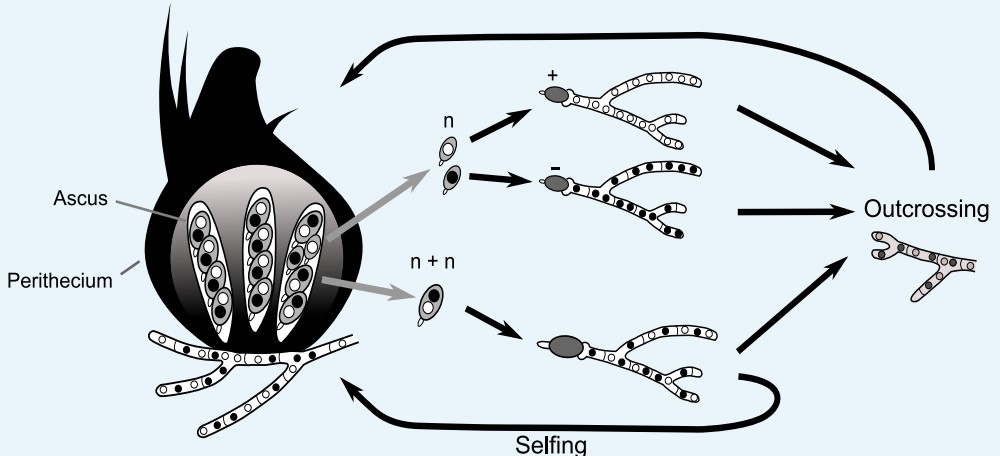

**Appendix 1—figure 1.** Simplified life cycle of *P. anserina* with explanation for isolating monokaryotic spores.

DOI: https://doi.org/10.7554/eLife.46454.040

### Two-locus spore-killing interaction

The interaction between *Psk-1* and *Psk-7* provides a good example of how the meiotic drive dynamics of *P. anserina* result in killing, even though both *Psk-1* and *Psk-7* possess the same *Spok* homologs (mutual resistance). The three *Spok* homologs (*Spok2*, *Spok3*, and *Spok4*) are all present in both *Psk-1* and *Psk-7*. The observed mutual resistance is thus due to the fact that the *Spok* block (with *Spok3* and *Spok4*) is located on different chromosomes. Because chromosomes segregate independently at meiosis, the expected killing percentage can be calculated as:

$$0.5^*\text{fk1}^*\text{fk2} = \text{fsk}$$

where 0.5 is due to independent assortment of chromosomes, fk1 is the killing percentage of strain 1, fk2 is the killing percentage of strain 2, and fsk is the spore-killing frequency observed between the two strains. For *Psk-1* crossed to *Psk-7*, this equals 0.27. This agrees well with the observed killing percentage of 23–27% (*Figure 4—figure supplement 1*).

## Appendix 2

DOI: https://doi.org/10.7554/eLife.46454.004

### History of spore-killer research in *Podospora*

Throughout the history of spore-killing research in *Podospora*, a number of observations have been made along with corresponding hypotheses. The discovery of *Spok3* and *Spok4* provides us with the opportunity to reinterpret these data in light of the results presented herein. Here, we will address data from four important studies: *Padieu and Bernet (1967)*, *van der Gaag et al. (2000)*, *van der Gaag et al. (2003)*, and *Hamann and Osiewacz (2004)*.

### Inconsistencies among the *Psk* designations

Our phenotyping is in accordance with the results of *van der Gaag et al. (2000)* for strains Wa28, Wa53, Wa58, Wa63, Wa87, S, and Z, but contradictions were observed for Wa21, Wa46, Wa47, and Y. Strain Wa21 was previously categorized as *Psk-3*, which is typified by inconsistent spore killing with *Psk-S* strains. Here we observed stable percentages and thus consider Wa21 to be representative of *Psk-2*. The role of *Psk-3* as a spore killer has been in doubt since its description (*van der Gaag et al., 2000*). This is in part due to the fact that the ascospores are not fully aborted as they are for the other spore-killer types. Instead, small transparent ascospores can still be observed within the ascus. Here, we were unable to find support for this spore-killer type and there is no clear correlation between its phenotype and any *Spok* genes. We therefore find it likely that the effect is due to other incompatibility factors rather than meiotic drive.

We did not observe any spore-killing in crosses between Wa46 (*Psk-4*) and Wa47 (*Psk-6*) as reported in *van der Gaag et al. (2000)*. Two other strains had been annotated as *Psk-6*, Wa89 and Wa90, but no other strains were recorded as *Psk-4*. Unfortunately we were not able to phenotype these strains and so we are unable to evaluate *Psk-6* further in this study. In addition, results from crosses of *Psk-4* with a *Psk-S* strain (Wa63) reveals that there is a dominance interaction between them, with *Psk-S* killing *Psk-4*. This is the opposite of what was proposed in *van der Gaag et al. (2000)*, i.e. that *Psk-S* kills *Psk-4*. Potentially, the original interpretation was hindered by poor mating of the *Psk-4* strain with tester *Psk-S* strains. Previously, strain Y was reported to have mutual resistance with *Psk-1*, to be susceptible to *Psk-7*, and to be dominant over all other types. Here we report that Y is susceptible to *Psk-1* and *Psk-7*, and has mutual killing with all other types, except in crosses with naïve strains where it is dominant.

### Allorecognition (*het*) genes and spore killing

As the *het-s* gene is capable of causing both vegetative incompatibility and spore killing, it was hypothesized that the *Psk* loci may be as well. The $S_5$ strains all demonstrate barrage formation (symptomatic of vegetative incompatibility) with strain S (*van der Gaag et al., 2003*). However when additional backcrosses were performed to generate $S_{14}$ strains, no barrages were observed (*Appendix 2—figure 1*). This indicates that the spore-killing types do not directly affect vegetative incompatibility or vice versa, but may be linked to loci which do. Note that the $S_5$ strains contain multiple genomic regions that are not isogenic with S, some of which contain known allorecognition genes (*Figure 4—figure supplement 3*).

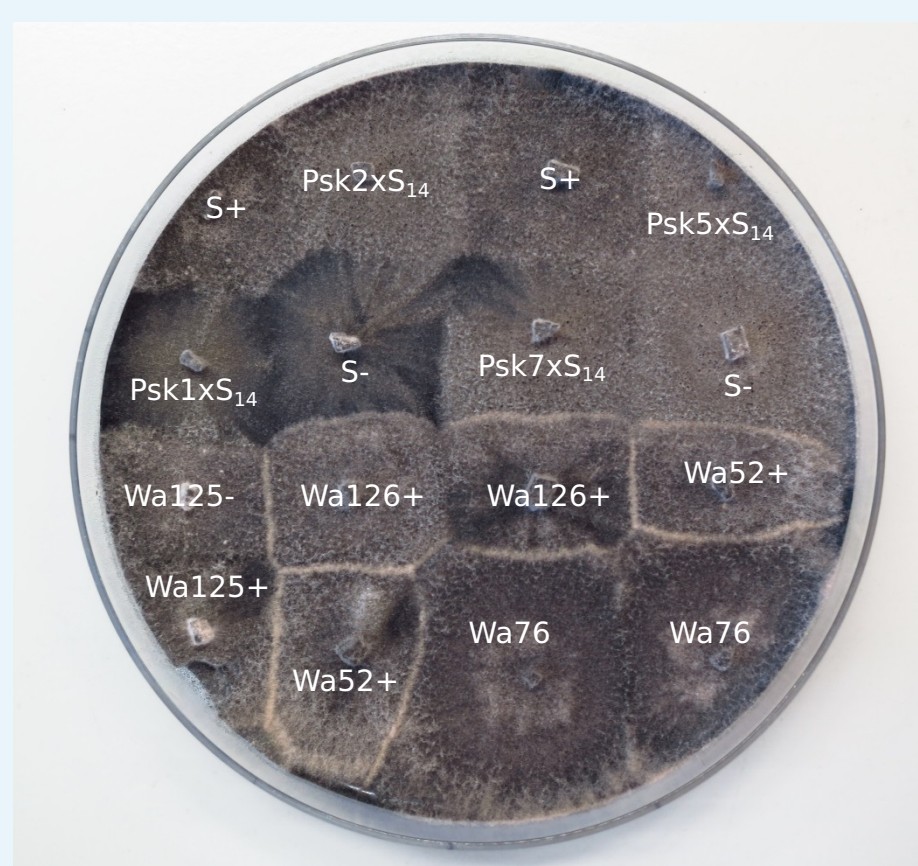

**Appendix 2—figure 1.** Barrage tests of the $S_{14}$ strains. Strains Wa126, Wa76, Wa52, and Wa125 are wild isolates of *P. anserina* in the Wageningen collection. The thick white lines of mycelia demonstrate a barrage, which is indicative of heterokaryotic incompatibility in fungi. No barrages are seen among the $S_{14}$ strains.

DOI: https://doi.org/10.7554/eLife.46454.042

## Incomplete penetrance of *Spok2*

To investigate the nature of the three-spored asci, tetrad dissections were conducted with asci from crosses between the *Psk-S* strains Wa63 and Us5, and the naïve strain Wa46. If the three-spored asci were the result of a four-spored ascus in which one of the spores aborted, all three spores should be heteroallelic for *Spok2*. If the three-spored asci are the result of incomplete penetrance of the killing factor, two spores should be homoallelic for *Spok2* while the other spore should have no copy of *Spok2*. Unfortunately, spores from the crosses had very low germination rates (1/15 for Wa63 x Wa46 and 1/12 for Us5 x Wa46) as compared to other crosses (generally close to 100% germination). The progeny from the successfully germinated spores were backcrossed to the parental strains and also allowed to self to infer their *Spok2* genotype. Crosses with the Wa63/Wa46 progeny revealed this strain to be homoallelic for *Spok2*, and crosses with the Us5/Wa46 progeny revealed it to have no copy of *Spok2*. Both of these observations are consistent with the hypothesis of incomplete penetrance of *Spok2*.

## Strain T and the original reports of spore killing in *Podospora*

The strain T has featured prominently in a number of important publications on spore killing in *Podospora*. It was one of the two strains investigated in the original description of spore killing by *Padieu and Bernet (1967)* translated and reinterpreted by *Turner and Perkins (1991)*, it was the strain in which *Spok1* was described (*Grognet et al., 2014*), and it was part of an investigation of spore killing in German strains of *Podospora* (*Hamann and Osiewacz, 2004*).

Our results clearly demonstrate that two strains labeled as T (T$_G$ and T$_D$ herein) are not only different strains, but are different species. The description of spore-killing in *Padieu and Bernet (1967)* matches our observations of crosses between T$_G$ and the *Psk-S* strain Wa63, including incomplete penetrance as implied by the presence of three-spored asci. Thus, we believe T$_G$ to be representative of the original T strain. In light of this, we reinterpret the results of both *Padieu and Bernet (1967)* and *Hamann and Osiewacz (2004)* as informed by the interactions of the *Spok* genes.

In *Padieu and Bernet (1967)*, a cross between two strains, T and T', is described. They identify two genes (one present in T and the other in T') that cause spore-killing and interact as mutual killers. The gene from T has a killing percentage of 90%, whereas the one from T' has a killing percentage of 40% and occasionally produces three-spored asci. This fits well with a cross of *Psk-5* and *Psk-S*, in which *Psk-5* kills at 90% and *Spok2* of the *Psk-S* strain kills at 40%, but has incomplete penetrance resulting in three-spored asci. Unfortunately, strain T' has to our knowledge not been maintained in any collections, so this cannot be confirmed experimentally. However, *Psk-S* strains are the most abundant phenotype from French, German, and Dutch populations (T' was isolated in France along with T) (*van der Gaag et al., 2000*; *Grognet et al., 2014*; *Hamann and Osiewacz, 2004*).

*Hamann and Osiewacz (2004)* presented a number of interesting observations. They reported a new spore-killer type, identified progeny that appeared to demonstrate gene conversion of the killer locus, and observed apparent recombinant spore-killer types. The study mostly centres around strain O, which they report to be of the same spore-killer type as T$_G$ and should thus be *Psk-5* given our results. As such, we suspect that their focal cross between O and Us5, a *Psk-S* strain, is the same as that in the *Padieu and Bernet (1967)* paper. We have independently confirmed that Us5 (kindly provided by A Hamann and H Osiewacz) is *Psk-S*, but strain O has not been maintained in any collection. *Hamann and Osiewacz (2004)* also state that strain He represents a new type of spore killer. However, with O classified as *Psk-5*, the interactions of He match that of a *Psk-1* strain. Furthermore, strain He exhibited no spore-killing with a *Psk-1* strain from Wageningen. From the cross of O and Us5, they identify a number of progeny with unexpected genotypes. They interpret these genotypes as evidence of both gene conversion and recombinant spore killer types. However, under a two-locus model of mutual killing, both effects can be explained by incomplete penetrance of *Spok2* (*Appendix 2—figure 2*). As the cross with Us5 showed a particularly high degree of anomalous results, it is possible that Us5 contains a unique allele of *Spok2* that is a particularly weak killer.

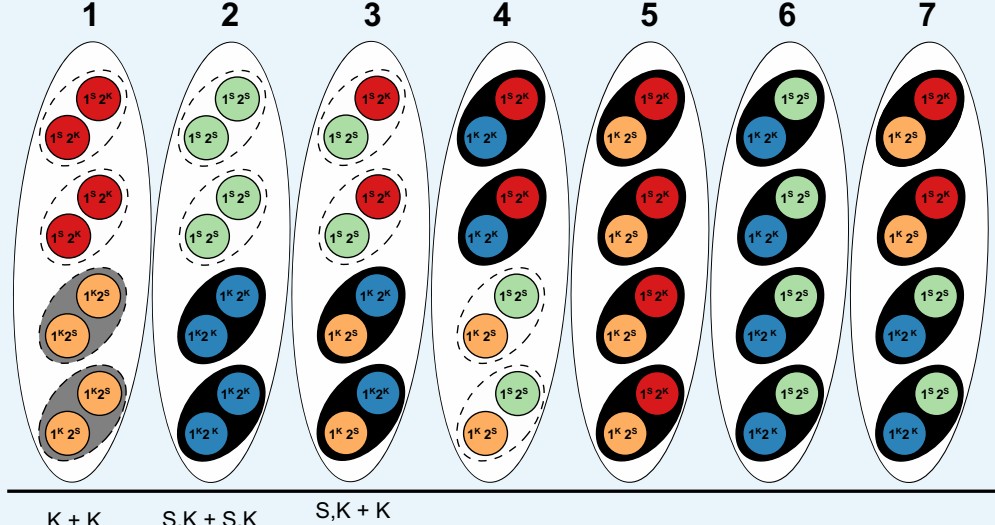

**Appendix 2—figure 2.** Explanation of results from a (presumably) *Psk-5* vs *Psk-S* cross in *Hamann and Osiewacz (2004)* with information about *Spok* genes as described in the text. The

seven asci represent the possible genotype combinations of a cross between a *Psk-5* strain and a *Psk-S*, as illustrated in **Turner and Perkins (1991)**. Black ovals represent the ascospores, dashed ovals represent killed spores, and colored circles represent the individual nuclei, with each color corresponding to a given genotype. Genotypes are annotated as in **Turner and Perkins (1991)**, wherein locus 1 corresponds to a killer locus with 90% FDS, the *Psk-5 Spok* block, and locus 2 represents a killer locus with 40% FDS, *Spok2*. Red nuclei represent the *Psk-S* parental genotype with *Spok2*, orange nuclei represent the *Psk-5* parental genotype with *Spok3* and *Spok4*, green nuclei represent the recombinant genotype with no *Spok* genes, and blue nuclei represent the recombinant genotype with *Spok2*, *Spok3*, and *Spok4*. Note that *Spok3* and *Spok4* are linked and do not segregate independently. Given its *Spok* content, the ascus type 1 should experience mutual killing, resulting in the abortion of all spores. Ascus types 2 to 4 should contain only two surviving spores, whereas ascus types 5 to 7 should have all spores surviving. Below, the asci are our interpretations of the annotations from **Hamann and Osiewacz (2004)**. In their study, these authors isolated F1 spores from four-spore asci, genotyped them on the basis of crosses to the parental strains, and found no spore-killing after selfing in a few cases. In their terminology, K + K strains would correspond to a strain with the *Psk-5* (grey spore) parental genotype of ascus type 1. This type of ascus should produce empty asci, so the fact that they are observed from four-spored asci suggests that when mutual killing occurs, the four spores may still develop. However, as no S + S strains (i.e. spores with two red nuclei in the type ascus 1) were reported, we can infer that only the *Psk-5* type (grey spore) may be viable. S,K + S,K strains are not indicative of a recombinant killer locus as suggested in the original work, but represent strains with all three *Spok* genes as produced in ascus type 2. The FDS frequencies reported suggest that the isolated strains are indicative of the blue nuclear genotype and not the green nuclear genotype. Hence, the spores containing only green nuclei might not be viable either. The S,K + K and K + S,K strains are indistinguishable from each other and are indicative of the surviving spores of a type 3 ascus. These strains should exhibit spore killing when selfed because of the distribution of *Spok2*. However, cases in which spore killing was not observed in their study could be explained by incomplete penetrance of *Spok2*. In all cases, these strains should not have been isolated from four-spored asci, indicating that either methodological issues occurred or that spore killing may still produce four-spored asci, but where the spores which should be absent are instead inviable.

DOI: https://doi.org/10.7554/eLife.46454.043

