## [Decision Letter]

Thank you for submitting your article "Combinations of *Spok* genes create multiple meiotic drivers in *Podospora*" for consideration by *eLife*. Your article has been reviewed by three peer reviewers, one of whom is a member of our Board of Reviewing Editors, and the evaluation has been overseen by Diethard Tautz as the Senior Editor. The following individual involved in review of your submission has agreed to reveal their identity: Sarah E Zanders (Reviewer #2).

The reviewers have discussed the reviews with one another and the Reviewing Editor has drafted this decision to help you prepare a revised submission.

Summary:

The manuscript by Vogan and colleagues is focused on the classical spore killing system of meiotic drive in the filamentous fungus *Podospora* anserina. Using an impressive combination of genetics, genomics, and molecular evolution approaches, the work greatly extends what is known about meiotic drivers in *Podospora*. Vogan et al. conclude that the *Psk* spore killing phenotype is due to alleles of previously identified *Spok* genes. The authors extend the analysis of *Spok* gene evolution and find that the *Spok3* and *Spok4* genes have moved around the genome in different lineages as part of a *Spok* sequence block. In addition, the authors significantly extend what is known about the potential molecular mechanisms of the *Spok* genes, a broadly important finding. Overall, this study is an important contribution that addresses fundamental questions related to the functioning and evolution of selfish genomic elements. *P. anserina* presents and excellent model system as it can be easily crossed, genetically manipulated and spore killing easily identified. Moreover, spore killing in *P. anserina* is a classic text-book example in genetics, and the present study provides novel insight into the underlying molecular mechanisms of a fungal meiotic drive system.

Essential revisions:

1) This is a wonderful piece of work but its value is at present reduced by the lack of clarity in presentation and writing. All reviewers noted that the experiments and results are not well presented and made numerous suggestions as to how they could be improved. The manuscript is extremely long and includes many details, which would be better summarized in a well-written supplementary text. The appendix includes a mini-review on *Podospora* genetics and biology although the main text already includes a box that summarizes necessary information on meiosis in *Podospora*. A more clear presentation of experiments conducted here as well as appropriate citation for previous studies could replace the "mini-review" in the appendix. Perhaps a figure that illustrates the different crossing experiments and the spore killing phenotypes would be helpful. We suggest substantial re-writing and careful consideration of the presentation of results and experimental set up.

2) The d_N_/d_S_ molecular evolutionary tests, where the authors conduct pairwise comparisons of the different *spok* genes, require revision. For example, it is not clear why the authors resorted to this approach when they could examine d_N_/d_S_ changes over the phylogeny of *spok* genes, since tests using the entire tree of *spoks* are going to be much more accurate. Furthermore, given the inference that *spok* genes sometimes undergo gene conversion, it is not clear how one is to interpret d_N_/d_S_ ratios or the phylogeny of *spok* genes in general (since the effect of these conversions on the phylogeny is to erase it). Finally, the M1 and M2 models applied here are for inference of selection based on interspecific data but details about these analyses were not provided in the Materials and methods part (you could provide the output of the PAML analyses in the supplementary material as a table with the d_N_/d_S_ estimates and likelihood ratio test). More clarity on this aspect of the authors' work is necessary.

3) The authors' inference that *spok* genes underwent cross-species transfer (last line of Abstract) is tenuous and the available data for this inference are very thin. It is fine to have it as "speculation" in the Discussion section, but we think this claim needs to be removed from the Abstract. The authors seem to invoke this transfer to explain the presence of *spok1* in *P. comata* and *spok4* in *P. anserina*. Given that the two genes have the same genetic structure, it makes sense they share a common ancestor. But what is not clear is why it is not possible that the common ancestor of these two species had a gene, which after their splitting became *spok1* in one species and *spok4* in the other. Or are we missing something?

4) A large part of the study relies on analyses of the population genomic dataset. Nevertheless, there is a dearth of information on the comparison of genomes, including regions outside of the *Spok*-encoding regions. Also the content and distribution of transposable elements which are proposed to support the diversification of *Spok* genes. The *Spok* block is found at different locations in the genome of different isolates. To assess if synteny breaks and frequently occurring rearrangements on small genome scales are particular traits that characterize this "selfish" block, more details on overall genome synteny would be informative.

5) Considering evolution of the *Spok* genes in *P. anserina* populations, an important question is whether the *Spok* genes confers a fitness cost during the life cycle of *P. anserinia*. This aspect is poorly addressed although the knock-in and knock-out strains of *Spok3* and *Spok4* should allow the authors to assess this.

---

## [Author Response]

Essential revisions:1) This is a wonderful piece of work but its value is at present reduced by the lack of clarity in presentation and writing. All reviewers noted that the experiments and results are not well presented and made numerous suggestions as to how they could be improved. The manuscript is extremely long and includes many details, which would be better summarized in a well-written supplementary text. The appendix includes a mini-review on Podospora genetics and biology although the main text already includes a box that summarizes necessary information on meiosis in Podospora. A more clear presentation of experiments conducted here as well as appropriate citation for previous studies could replace the "mini-review" in the appendix. Perhaps a figure that illustrates the different crossing experiments and the spore killing phenotypes would be helpful. We suggest substantial re-writing and careful consideration of the presentation of results and experimental set up.

We would like to thank the reviewers’ many comments on how to improve the presentation of the results. We have done our best to follow them and have rearranged the first half of the Results section accordingly. We minimized the amount of detail presented in favour of more general summaries. We still would like to keep the mini-review, as some of the reviewers had issues understanding the sequencing of the monokaryons (i.e. the life cycle is also important, not only the meiosis) and have attempted to integrate references to it better when applicable. We have added the recommended figure as Figure 5—figure supplement 4, and included more details of the crossing designs in the Materials and methods.

2) The dN/dS molecular evolutionary tests, where the authors conduct pairwise comparisons of the different spok genes, require revision. For example, it is not clear why the authors resorted to this approach when they could examine dN/dS changes over the phylogeny of spok genes, since tests using the entire tree of spoks are going to be much more accurate. Furthermore, given the inference that spok genes sometimes undergo gene conversion, it is not clear how one is to interpret dN/dS ratios or the phylogeny of spok genes in general (since the effect of these conversions on the phylogeny is to erase it). Finally, the M1 and M2 models applied here are for inference of selection based on interspecific data but details about these analyses were not provided in the Materials and methods part (you could provide the output of the PAML analyses in the supplementary material as a table with the dN/dS estimates and likelihood ratio test). More clarity on this aspect of the authors' work is necessary.

This point made us aware that we should remove the PAML-analysis from the study. The reason is that, after consideration, we think the PAML analysis is not suitable for our dataset as the relationship of the *Spok* genes appears very complex, including possible cross-species transfers and gene conversions. Given this complexity, we risk violating the assumptions of the analyses and therefore prefer to remove it. However, we have kept the table with amino acid differences and d_N_/d_S_for each pairwise comparison of the *Spok*s, but moved it from the section on functional annotation to identification of novel *Spok* genes.

Analysing the evolution of the *Spoks* across the broader fungal phylogeny would address a different question then our aim here. It is unknown whether homologs in other species function as meiotic drive genes or if they have also been transferred horizontally. As such it is difficult to construct valid hypotheses to test with the broader data set and we again run the risk of violating the assumptions of the model, so we have opted to not conduct this analysis.

3) The authors' inference that spok genes underwent cross-species transfer (last line of Abstract) is tenuous and the available data for this inference are very thin. It is fine to have it as "speculation" in the Discussion section, but we think this claim needs to be removed from the Abstract. The authors seem to invoke this transfer to explain the presence of spok1 in P. comata and spok4 in P. anserina. Given that the two genes have the same genetic structure, it makes sense they share a common ancestor. But what is not clear is why it is not possible that the common ancestor of these two species had a gene, which after their splitting became spok1 in one species and spok4 in the other. Or are we missing something?

We agree with the reviewer that the results pointing towards cross-species transfer could be better and more carefully disentangled and discussed. Accordingly, in the Discussion, we have made a more explicit description of different scenarios, and added a figure to illustrate these (Figure 8). Our conclusion on cross-species transfer is based on two major points. First, the *Spok* genes from different species have reticulated relationships, with multiple gene features (indels and repeat motifs) shared between *Spok1* and *Spok4*. This pattern is in direct contradiction with the structure of the block that suggests *Spok3* and *Spok4* are paralogs produced from tandem duplications. The discussion of the new figure addresses this contrast extensively. The second point involves the phylogeny of closely related SPOK homologs across fungi, which places homologs from very different taxonomic classes close together. We hope that we have made the case more clear and that you think it is valid to say in the Abstract that “Genomic and phylogenetic analyses across ascomycetes suggest that the *Spok* genes disperse via cross-species transfer, and evolve by duplication and diversification within several lineages.”

4) A large part of the study relies on analyses of the population genomic dataset. Nevertheless, there is a dearth of information on the comparison of genomes, including regions outside of the Spok-encoding regions. Also the content and distribution of transposable elements which are proposed to support the diversification of Spok genes. The Spok block is found at different locations in the genome of different isolates. To assess if synteny breaks and frequently occurring rearrangements on small genome scales are particular traits that characterize this "selfish" block, more details on overall genome synteny would be informative.

We agree with the reviewer on this point and accordingly, we have added two new analyses illustrated by two new figures (Figure 1 and Figure 1—figure supplement 1). We have in the beginning of the Results added the section “The *Podospora* species are closely related and highly syntenic”, in which we show (1) species divergence/diversity, and (2) a circos plot with collinear regions between species. A new supplementary figure shows inversions as well. Furthermore, we added another supplementary figure (Figure 4—figure supplement 1) with the alignment of chromosome 5 for three representative strains. This plot serves the double purpose of showing overall collinearity interrupted by the *Spok* block, and to show the small scale movement of transposable elements (which are correlated to low GC content in *Podospora*). It also provides context for the position of *Spok2* and *SpokΨ1*.

5) Considering evolution of the Spok genes in P. anserina populations, an important question is whether the Spok genes confers a fitness cost during the life cycle of P. anserinia. This aspect is poorly addressed although the knock-in and knock-out strains of Spok3 and Spok4 should allow the authors to assess this.

We agree that this is very important for the understanding of drive and *Spoks*, but we argue not to have it in this particular paper since 1) it is not really the scope and 2) it needs its large attention (measuring fitness in *Podospora* is not trivial).